# Learning Interactive World Model for Object-Centric Reinforcement Learning

**Fan Feng**[1,2]     **Phillip Lippe**[3*]     **Sara Magliacane**[3]

[1] University of California San Diego [2] Mohamed bin Zayed University of Artificial Intelligence
[3] University of Amsterdam

{ffeng1017,phillip.lippe,sara.magliacane}@gmail.com

## Abstract

Agents that understand objects and their interactions can learn policies that are more robust and transferable. However, most object-centric RL methods factor state by individual objects while leaving interactions implicit. We introduce the Factored Interactive Object-Centric World Model (FIOC-WM), a unified framework that learns structured representations of both objects and their interactions within a world model. FIOC-WM captures environment dynamics with disentangled and modular representations of object interactions, improving sample efficiency and generalization for policy learning. Concretely, FIOC-WM first learns object-centric latents and an interaction structure directly from pixels, leveraging pre-trained vision encoders. The learned world model then decomposes tasks into composable interaction primitives, and a hierarchical policy is trained on top: a high level selects the type and order of interactions, while a low level executes them. On simulated robotic and embodied-AI benchmarks, FIOC-WM improves policy-learning sample efficiency and generalization over world-model baselines, indicating that explicit, modular interaction learning is crucial for robust control[2].

## 1 Introduction

World models aim to learn state abstractions and action-conditioned dynamics that capture the evolution of high-dimensional observations, along with auxiliary information (e.g., rewards, skills), for decision-making tasks [1–5]. Recent advances have demonstrated their effectiveness in downstream applications, such as robotics [2, 6–10] and autonomous driving [11–14].

One of the central challenges in world model is to extract low-dimensional, structured latent representations from high-dimensional observations, which often display high complexity and variability across both semantic and dynamic aspects. On the dynamics side, latent spaces often contain underlying structures [15, 16]. Prior work imposes structural priors to learn compact latents that encourage disentanglement and capture relational or compositional patterns [17–23]. On the semantics side, pre-trained visual features are leveraged to better encode rich content and improve fidelity [9, 24–31]. Collectively, these approaches learn compressed, structured representations of high-dimensional perceptual data to support downstream decision making. However, it remains unclear to what extent such compression and structure are necessary and sufficient for down-streaming policy learning.

In this work, we study which types and degrees of decomposition structure make latent representations effective for efficient and generalizable policy learning. Real-world settings exhibit substantial variability in both visual appearance and dynamic interactions, often involving multiple objects with diverse attributes. It is therefore natural to reason in terms of *objects*, their *interactions*, and

---

[*]Now at Google Deepmind

[2]Project page: https://sites.google.com/view/fioc-wm.

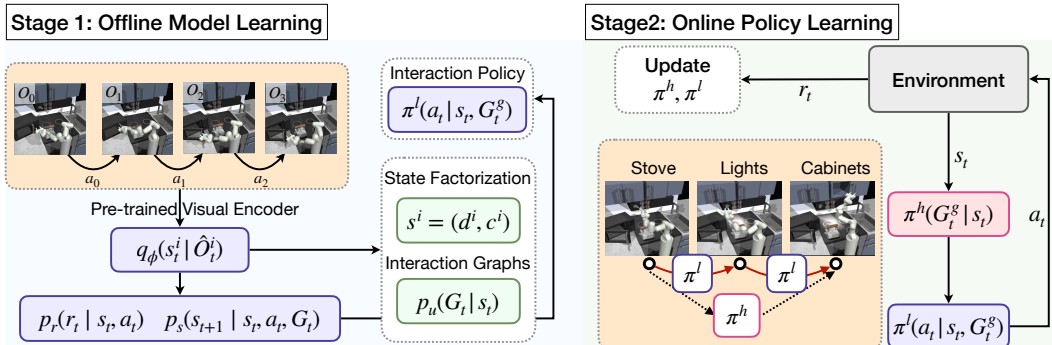

Figure 1: The overall pipeline, including offline model learning (left) and online policy learning (right) phases. The illustrative examples are from the Franka-kitchen environment [40].

the *attributes that induce these interactions*. To this end, we propose the Factored Interactive Object-Centric World Model (FIOC-WM), which learns a two-level factorization: an *object-level* representation with explicit interactions, and an *attribute-level* representation for each object. This factorization is then exploited for down-streaming planning and control.

At the *object* level, we consider both the decomposition of scenes into independently evolving objects and the modeling of their interactions. Modeling the interactions among objects is crucial for effective policy learning as real-world dynamics are heavily influenced by rich interactions among objects, such as collisions, containment, stacking, and physical forces like friction or gravity, which collectively determine the evolution of the environment [32–34].

At the *attribute* level, each object can be further factorized into attributes based on their temporal behavior, e.g., if they are static (e.g., color, shape) or dynamic (e.g., position, velocity) over time. This factorization provides a principled inductive bias to reduce redundancy and highlight the minimal sufficient components needed for planning and control. Importantly, this also supplements the accurate object-level interaction modeling as the interaction can be further factorized: for each object, only the dynamic part (e.g., position, velocities) will be changed during interactions with others. By incorporating both object-level and attribute-level factorization, we can precisely model the dynamics of all objects, including their interactions.

This structured modeling enables accurate prediction of system behavior and allows the learned interaction models to serve as efficient surrogates for decision making. Building on recent hierarchical RL with object-centric subgoals [35–37], we instantiate subgoals as object interactions, allowing complex tasks to be decomposed into sequences of interaction primitives and thereby enabling more efficient planning and control.

FIOC-WM jointly factorizes the static attributes and dynamic variables of each object in the environment, as well as their interactions with each other and the agent. After learning the FIOC-WM, we can then leverage its interaction models to learn an interaction-centric policy. This enables efficient solutions for long-horizon policy learning. Inspired by recent work [26, 27], we use pre-trained visual embeddings [38, 39] as surrogates for raw high-dimensional observations, facilitating the learning of semantically meaningful latents. FIOC-WM can recover interactions and learn the factorized states within the latent representations derived from these visual embeddings. The learned interactions are then used to train a policy designed to induce the desired interactions between objects. These offline-learned policies are subsequently employed as composable modules for long-horizon tasks. We evaluate FIOC-WM on a diverse set of robotic control and embodied AI benchmarks, demonstrating enhanced world model capability and more efficient downstream policy learning by employing the appropriate factorization and leveraging it as sub-tasks.

## 2 Factored Interactive Object-centric POMDP

We focus on a Partially Observable Markov Decision Process (POMDP) [41] and consider an environment in which objects interact with each other, and in which there are global latent factors that can affect or modulate these interactions. We denote the state at timestep $t$ as $\mathbf{s}_t$ and assume it can be

factored across $N$ objects. Moreover, we assume that the state of each object $i$ can be represented as $\mathbf{s}_t^i = \{\mathbf{d}_t^i, \mathbf{c}^i\}$, where $\mathbf{d}_t^i$ represents the dynamic, time-varying variables (e.g., position, velocity) and $\mathbf{c}^i$ represents the constant, time-invariant properties such as color, mass, and friction, some of which can affect the dynamics of the object.

We represent interactions between objects with a sequence of time-varying graphs $\mathcal{G} = \{G_1, \ldots, G_T\}$, where each edge in a graph $G_t$ captures an interaction between two objects at time $t$. This models that at each timestep, different objects might interact. We also assume that these graphs are sparse, meaning that at each timestep there are only a subset of objects interacting.

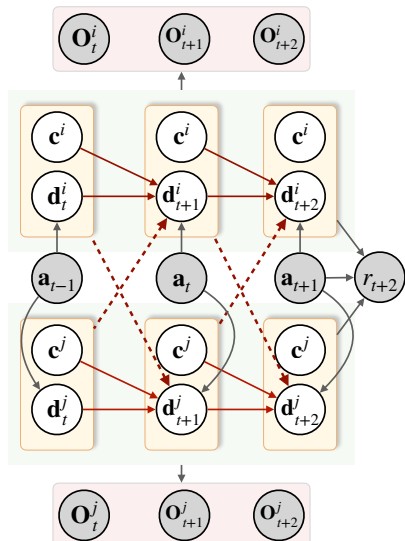

For each object, we define a self-transition function $f_{\text{self}}$, which represents the evolution of the object dynamics without interactions. In the self-transition function the constant properties influence the evolution of its dynamic variables over time, but not viceversa. When two objects $i$ and $j$ interact, an object can only affect the dynamic variables of the other object through the interaction transition function $f_{\text{inter}}$. More formally, we model that the state transition for object $i$ follows the form:

$$\mathbf{d}_{t+1}^i = f_{\text{self}}(\mathbf{d}_t^i, \mathbf{c}^i, \mathbf{a}_t, \epsilon_t) + \sum_{j \in \mathcal{N}_t(i)} f_{\text{inter}}(\mathbf{d}_t^i, \mathbf{d}_t^j, \mathbf{c}^j, \delta_t),$$

where $\mathcal{N}_t(i)$ denotes the set of objects interacting with object $i$ at time $t$, and $\epsilon_t$ and $\delta_t$ indicate the latent noise variables that model the stochasticity of the system.

We assume that also the observations $\mathbf{o}_t$ are factored across the $N$ objects and that the generating process for observation of object $i$ at time $t$ is $\mathbf{o}_t^i = g(\mathbf{s}_t^i, \epsilon_t^i)$, where $\epsilon_t^i$ is a latent i.i.d. random noise that represents the stochasticity in the observations. Finally, as in standard settings, the reward function is a function of the global state $\mathbf{s}_t$ and the action, i.e., $r_t = h(\mathbf{s}_t, \mathbf{a}_t)$.

Figure 2: An example of a FIOC-POMDP, where we only show the reward for $t + 2$ for clarity. Gray nodes are observed variables, while white nodes are latent variables. Each orange box represents the state of an object (in this case, objects $i$ and $j$). Red solid edges are the state transition per object, and dashed edges are the interactions among objects.

We call a model that satisfies all of these assumptions a Factored Interactive Object-centric POMDP (FIOC-POMDP). Fig. 2 depicts an example of a FIOC-POMDP.

## 3 Learning the FIOC World Models

The overall framework (Fig. 1) consists of two stages: (1) offline model learning (Fig. 3) and (2) hierarchical policy learning. In offline model learning, we learn a world model for a FIOC-POMDP as two-level factorization of the latent space, at the object and attribute levels, and model latent dynamics based on object interactions. Leveraging the learned interactions, we train an inverse dynamics model to map the states of two separate objects to the states where they interact effectively, which we use as an interaction policy. In hierarchical policy learning, a hierarchical policy is trained. The high-level policy selects a sequence of target interaction graphs, while the low-level interaction policy trained in the first stage executes them by inducing the corresponding interaction graph in the environment.

### 3.1 Stage 1: Offline Model Learning

We encode the observations using object-centric representation learning built on top of pre-trained models such as DINO-v2 [38] and R3M [39], which have been empirically shown to provide high-quality image understanding capabilities [42–44, 27, 45] and facilitate robotic manipulation tasks [25, 26, 31]. Building on the empirical and theoretical work regarding the recoverability of latent features from supervised pre-trained models [46], we assume that these embeddings provide sufficient features and information for world models. This includes supporting the dynamics and reward models, as well as capturing action-related features effectively. Then, similarly to Zadaianchuk

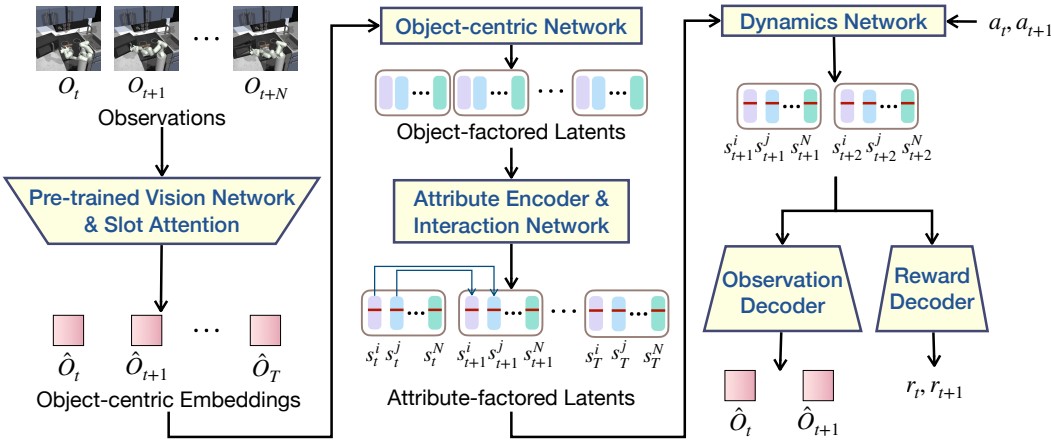

Figure 3: The pipeline of Offline Model Learning (Stage 1) jointly learns the observation function, state factorization, dynamics model, and reward model. Although Fig. 2 includes low-level policy learning as part of Stage 1, for clarity, we defer the discussion of low-level policy learning to Stage 2.

et al. [47], we use slot attention [48] to cluster the object-centric representation on top of the embeddings. The slot attention outputs a set of slot representations, which we use as the factored observation $\{\hat{\mathbf{o}}^1, \hat{\mathbf{o}}^2, \ldots, \hat{\mathbf{o}}^N\}$ corresponding to the factored raw observation $\{\mathbf{o}^1, \mathbf{o}^2, \ldots, \mathbf{o}^N\}$. To map these factored observations to factored states $\{\mathbf{s}^1, \mathbf{s}^2, \ldots, \mathbf{s}^N\}$, we train a variational auto-encoder (VAE) [49] with the encoder $q_\phi(\mathbf{s}^i | \hat{\mathbf{o}}^i)$ and the decoder $p_\psi(\hat{\mathbf{o}}^i | \mathbf{s}^i)$, where $\mathbf{s}^i$ is the latent state corresponding to the observation $\hat{\mathbf{o}}^i$, and the shared parameters are used across all slots.

To encourage structured representations, we learn to factorize the latent state into static and dynamic components, denoted by $\mathbf{c}$ and $\mathbf{d}$, respectively. Two separate encoders, $f_c(\mathbf{s})$ and $f_d(\mathbf{s})$, are used to extract static and dynamic features from observations. We assume that static features remain invariant over time, while dynamic features evolve. To enforce this, we regularize the output of $f_c(\mathbf{s})$ to remain temporally consistent for each of the $N$ object slots:

$$\mathcal{L}_{\text{static}} = \sum_{t=1}^{T-1} \sum_{i=1}^{N} \left| f_c(\mathbf{s}_{t+1}^i) - f_c(\mathbf{s}_t^i) \right|^2, \tag{1}$$

where $T$ is the number of time steps. To ensure that different objects encode distinct static attributes, we use a contrastive loss [50] that separates static features across slots:

$$\mathcal{L}_{\text{con}} = -\sum_{t=1}^{T-1} \sum_{i=1}^{N} \log \frac{g\left(f_c(\mathbf{s}_t^i), f_c(\mathbf{s}_{t'}^i)\right)}{g\left(f_c(\mathbf{s}_t^i), f_c(\mathbf{s}_{t'}^i)\right) + \sum_{j \in \mathcal{N}} g\left(f_c(\mathbf{s}_t^i), f_c(\mathbf{s}_{t'}^j)\right)}, \tag{2}$$

where $t'$ is a different time step and $\mathcal{N}$ denotes a set of negative slots $j \neq i$ from the same scene. $g$ is the distance measurement of the representation, we use cosine similarity here.

For the dynamic features, we leverage their temporal evolution to model latent state transitions, as only $\mathbf{d}$ varies over time. We adopt the variational inference framework [49] to learn the encoder $f_d(\mathbf{s})$, parameterized by a GRU [51], which captures the dynamics of each object slot. The prior over the dynamic state $\mathbf{d}_t$ is factorized across the $N$ object slots as: $p_s(\mathbf{d}_t \mid \mathbf{d}_{t-1}, \mathbf{a}_{t-1}) = \prod_{i=1}^{N} p_s(\mathbf{d}_t^i \mid \mathbf{d}_{t-1}, \mathbf{a}_{t-1}, G_t)$, where $G_t$ denotes the interaction graph representing the relational structure among objects at time $t$. In other words, $G_t$ captures the pairwise interactions between objects at time step $t$, where each edge indicates whether an interaction exists between a pair of objects. Concretely, this is represented as a binary adjacency matrix of size $N \times N$, where $N$ is the number of objects.

The posterior over $\mathbf{s}_t$ is conditioned on the current the visual embeddings $\hat{\mathbf{o}}_t$ and the hidden state $\mathbf{h}_t = \text{GRU}(\mathbf{s}_{t-1}, \mathbf{h}_{t-1})$, as: $q_\phi(\mathbf{s}_t \mid \hat{\mathbf{o}}_t, \mathbf{h}_t)$. Then we use an observation decoder to reconstruct observations: $p_\sigma(\hat{\mathbf{o}}_t \mid \mathbf{s}_t)$, with the reconstruction loss:

$$\mathcal{L}_{\text{recon}} = \sum_{t=1}^{T} \left\| \hat{\mathbf{o}}_t - \hat{\mathbf{o}}_t^{\text{decoded}} \right\|^2, \tag{3}$$

where $\hat{\mathbf{o}}_t^{\text{decoded}}$ is sampled from $p_\sigma(\cdot \mid \mathbf{s}_t)$. To capture temporal consistency, we also predict the next-step observation:

$$\mathcal{L}_{\text{pred}} = \sum_{t=1}^{T} \left\| \hat{\mathbf{o}}_{t+1} - \hat{\mathbf{o}}_{t+1}^{\text{decoded}} \right\|^2 . \tag{4}$$

We encourage alignment between the posterior and the prior using KL divergence:

$$\mathcal{L}_{\text{KL}} = \sum_{t=1}^{T} \text{KL} \left( q_\phi(\mathbf{s}_t \mid \hat{\mathbf{o}}_t, \mathbf{h}_t) \,\|\, p_s(\mathbf{s}_t \mid \mathbf{s}_{t-1}^s, \mathbf{a}_{t-1}, G_t) \right) . \tag{5}$$

Similarly, we apply a reward decoder $p_r(r_t \mid \mathbf{s}_t, \mathbf{a}_t)$ based on the learned latent states and actions. The reward loss is as follows:

$$\mathcal{L}_{\text{rew}} = \sum_{t=1}^{T} \left\| \hat{r}_t - r_t \right\|^2 . \tag{6}$$

To learn the interaction graph $G_t$, we use the current estimated latent states $\mathbf{s}_t$ as input. We introduce a surrogate latent variable $\mathbf{u}_t$ that parameterizes the distribution over interaction graphs. This captures the underlying interactions that may vary over time.

Specifically, for each object pair $(i, j)$ at time $t$, we encode their latent states $s_t^i$ and $s_t^j$ using a GRU encoder to obtain a pairwise embedding:

$$\mathbf{u}_t^{ij} = f_{\text{enc}, \phi_u}(\mathbf{s}_t^i, \mathbf{s}_t^j) \tag{7}$$

The transition of $\mathbf{u}_t$ is modeled as: $p_u(\mathbf{u}_t \mid \mathbf{s}_t) = f_u(\mathbf{s}_t)$, where $f_u$ is a parameterized function that captures the dependencies among the current latent states $\mathbf{s}_t$. We consider two approaches for learning the state transition distribution $p_s$: (i) learning variational masks, following [52, 53]; and (ii) applying conditional independence testing, following [54]. The detailed loss functions are provided in Appendix C.2.

## 3.2 Stage 2: Online Hierarchical Policy Learning

In this section, we describe how we use the learned interactive world model for object-centric RL, particularly for long-horizon task learning. Our framework is built on the recent work that models the object interactions as skills [37]. The key intuition is that long-horizon tasks can be decomposed into a sequence of interactions.

Our approach first focuses on learning a *low-level policy* capable of invoking the desired interactions. Based on the learned interactive world model, we can accurately predict the dynamics of interactions and the regimes governing these interactions. This enables the agent to learn the policy by leveraging the predicted interactions to learn the inverse mapping from interactions to actions. We learn the low-level policy $\pi^l$ by employing model predictive control (MPC) [55, 56, 5] or proximal policy optimization (PPO) [57], where the initial and target interaction of two objects are provided. At time step $t$, we are given the target interaction graph at future steps from high-level policy, denoted as $G_t^g$, and the low-level policy is $\pi^l(\mathbf{a}_t \mid \mathbf{s}_t, G_t^g)$. Given the learned transition models $p_s$ and $p_u$, we use $\mathbf{s}^i$ and $\mathbf{u}^g$ to infer the target states $\mathbf{s}_g^i$ and $\mathbf{s}_g^j$. Using these inferred target states, we apply MPC or PPO to generate a sequence of actions that transitions the system from $t$ to $t + k$ while minimizing the discrepancy between the predicted and target states. We learn the low-level policy during world model learning (Stage 1), and then fine-tune it with online data during Stage 2, where the policy is updated each time new interaction data becomes available.

We then learn a schedule of interactions for the model to handle long-horizon tasks by optimizing the task reward. We learn the chain-of-interactions for the high-level policy $\pi^h : \mathcal{S} \to \mathcal{G}$, which selects the interaction graph $\mathcal{G}$ based on the input state $\mathcal{S}$. This implies that the action space corresponds to graph selection, but this space can grow exponentially with the number of objects. To address this, following previous works on skill discovery with object interaction [58, 37], we impose constraints by limiting the number of objects considered at each time step. Following the graph selection policy introduced in [37], at any given time, we focus on a fixed subset of objects (smaller than 2), leveraging a diversity reward $r_{\text{div}}$ as a surrogate to make the selection process diverse. We define $r_{\text{div}} = 1/\sqrt{|G_{\text{visited}}|}$, where $|G_{\text{visited}}|$ is the number of graphs that have been visited in the past transitions. Then the high-level policy $\pi^h(G_t^g \mid \mathbf{s}_t)$ is updated with both the task reward $r_{\text{task}}$ and this diversity reward $r_{\text{div}}$.

### 3.2.1 Practical Implementation

We assume that each state $\mathbf{s}_t$ is associated with an interaction graph $G^t$, and the final task corresponds to reaching a desired target graph $G^g$. The high-level policy $\pi^h$ selects a sequence of intermediate *subgoal graphs* that gradually transform $G^t$ into $G^g$, where each subgoal graph differs from the previous one in only a single interaction. For example, in a task such as moving a kettle from the counter to the stove, the graph transitions involve first enabling an interaction between the arm and the kettle, followed by an interaction between the kettle and the stovetop.

To make the subgoal selection both tractable and structured, we do not sample directly from the full space of possible object interactions. Instead, at each decision point, we first identify a small subset of objects (typically one or two objects) as primary candidates for initiating interaction changes. These candidates define the anchor object(s) $i$, and we then select a target object $j$ conditioned on $i$ to form the proposed subgoal interaction $(i, j)$. This scheme reduces the combinatorial action space and leads to more localized graph transitions. Note that the selected subset does not constrain the interaction to only occur between these objects; rather, it defines a focused region of the graph for subgoal exploration.

## 4   Related Work

Our framework aims to uncover interactive and factored object-based representations of environments, so it is closely related to factored RL, particularly object-centric RL. Factored RL models the environment in terms of Factored Markov Decision Process [59], where the state of the Markov decision process is factored in state components and sparse relationships exist among state components, actions, and rewards. This factorization enables efficient policy solutions [60, 61]. A specific type of factorization, which we also adopt, is object-centric reinforcement learning [62], where states or observations are grouped into object-centric clusters. In object-centric RL, actions typically target only a subset of objects, and rewards are often associated with the states of specific objects or object subsets. This facilitates more structured and efficient decision-making.

Recent works on object-centric RL can be broadly categorized into two major directions: (i) learning object-centric representation and (ii) modeling the object relations and policy architectures for compositional generalization. For the first line of research, approaches focus on using object-centric representation learning techniques [48, 63–65] to extract meaningful object-level features from raw observations. These methods then learn object-centric policies directly from object-centric representations [66, 67, 62]. The second line of work develops object-centric policies by modeling object relations and policy structures, incorporating inductive biases in the state transition and policy networks. Methods include the use of graph neural networks [68], linear relational networks [69], self-attention, and deep sets [70]. The learned object-centric states and relational structures are then used to achieve compositional generalization in reinforcement learning [71, 18, 33, 72, 22, 73]. Our work combines ideas from both directions, especially related to the series of works [33, 71, 22, 67], which learn factored state attributes, providing a more fine-grained representation than object-centric factorizations, and also model the interactions among objects to achieve compositional generalization. However, we go beyond object-centric policies by learning an interaction-centric policy.

Our framework, which uses low-level and high-level policy for decomposing complex tasks into interaction learning, is similar to hierarchical reinforcement learning (HRL). HRL typically consists of a high-level policy (often referred to as an option [74], sub-skills, or sub-goals in the literature) and a low-level policy, enabling the efficient learning of complex RL tasks [75]. Within the scope of HRL, the most close to our work is the line of research that focuses on learning goal-conditioned hierarchical policies or hierarchical skill discovery. Zadaianchuk et al. [66] propose the goal-conditioned hierarchical policies with learning object-based hierarchical goals. Hierarchical skill discovery focuses on decomposing complex tasks into object-wise or object-interaction-based components. For instance, Wang et al. [37] use conditional independence testing to identify sub-goals, while Chuck et al. [35, 73] and Hu et al. [76] use Granger causality or counterfactual reasoning to uncover hierarchical structures. Our work is also built upon those works in using interactions as sub-skills [36, 37], but we learn interaction models jointly with observations and dynamics within the world model directly from high-dimensional inputs.

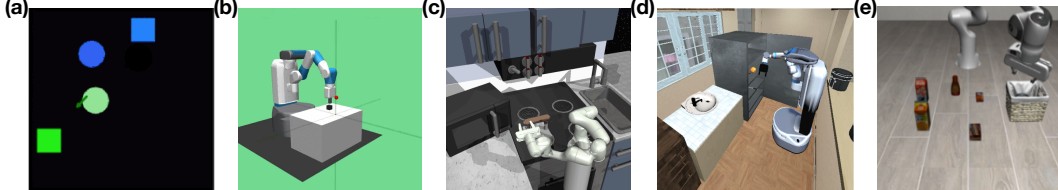

Figure 4: Visualization of evaluated benchmarks: (a). Sprites-World; (b). OpenAI-Gym Fetch; (c). Franka Kitchen; (d). i-Gibson; and (e). Libero. A larger version is in Table A4.

## 5  Experiments

To evaluate the effectiveness of our proposed interactive world model and policy learning framework, we aim to address the following questions: (i) *How accurately does the model learns the state disentanglement and interaction models?* (ii) *How well does it perform in long-horizon task learning?* and (iii) *How well does the framework achieve compositional generalization?*

To answer these questions, we evaluate our method on a range of simulated control, robotic manipulation, and embodied AI benchmarks, including SpritesWorld [77], OpenAI-Gym Fetch [78], iGibson [79], and Libero [80]. We consider both reinforcement learning and imitation learning tasks.

**Baselines.**   For online RL or imitation learning, we compare against established baselines including DreamerV3 [4], TD-MPC2 [81] and the object-centric model-free method EIT [67]. For offline RL, we compare with DINO-WM [27], which also leverages DINO-based pretraining for downstream planning.

**Benchmarks.**   We consider long-horizon tasks that require completing several sub-skills to achieve the overall objective. OpenAI Gym Fetch [78] is a simulated environment featuring a Fetch robotic arm capable of manipulating cubes and switches. The tasks involve completing sub-tasks that require pushing or switching a varying number of objects. Franka-kitchen [40] is an environment where the 7-DoF Franka Emika Panda arm performs tasks in a kitchen. We consider several sequential sub-tasks, such as turning on the microwave, moving the kettle, turning on the stove, and turning on the light. i-Gibson [82] is a simulated environment with a Fetch robot operating in everyday household tasks with rich objects and interactions. Similarly to [37], we consider the tasks with the peach object. Libero [80] is a benchmark for lifelong robot learning and imitation learning in household and tabletop environments. We focus on randomly selected tasks within libero-goal.

### 5.1  Evaluation Metrics.

In addition to evaluating policy learning and planning performance, we assess the effectiveness of world model learning by examining three key aspects: observation and dynamics modeling, interaction learning, and disentanglement quality. Specifically, for all methods (excluding those evaluated under nSHD), we adopt variational masks to infer the interaction structures. For downstream control, we apply MPC for Gym-Fetch and Franka-Kitchen, and use PPO for LIBERO and iGibson.

**Observation and Dynamics Modeling**   We measure the predictive quality of future observations using the Learned Perceptual Image Patch Similarity (LPIPS) metric [83], which evaluates perceptual similarity between predicted and ground-truth image patches.

**Interaction Learning**   We evaluate the ability of our model to learn interactions through two approaches: (i) *Variational mask learning*, where the state encodes adjacency matrices as latent variables. Each edge is sampled from either a differentiable approximation of a categorical distribution [84, 85, 52] or a discrete codebook [53]. (ii) *Conditional independence testing*, where we test for the existence of interaction using parametric models to predict dynamics [54]. We compare our approach with baselines that do not explicitly model dynamic structure, but instead rely on post hoc analysis based on attention weights to infer interactions, as in local causal discovery methods [86, 87]. We use normalized Structured Hamming Distance (nSHD). Further details are provided in Appendix C.2.

| Environment | Dreamer-V3 | TD-MPC2 | EIT | DINO-WM | FIOC |
|---|---|---|---|---|---|
| Fetch | 0.042 | 0.039 | 0.026 | 0.009 | **0.007** |
| Kitchen | 0.102 | 0.123 | 0.096 | **0.035** | 0.038 |
| Libero | 0.089 | 0.061 | 0.040 | 0.035 | **0.027** |

Table 1: Comparison of world models on LPIPS metrics on Fetch, Kitchen, and Libero.

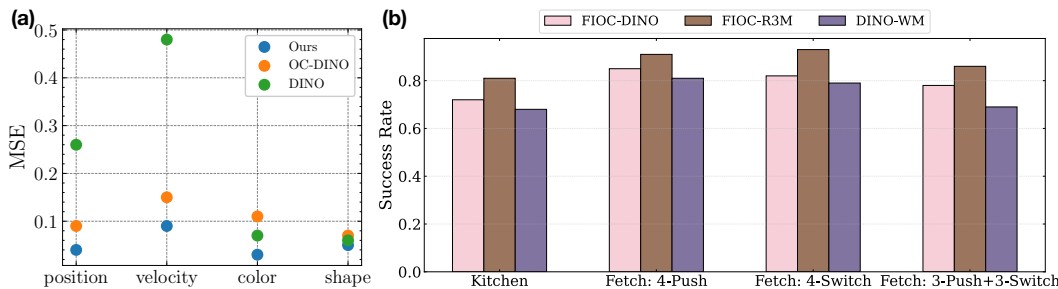

Figure 5: (a) Evaluation of state factorization in Sprites-world. We report MSE from linear probing to assess the quality of the learned representations against ground-truth attributes.(b) Offline RL performance (success rate) comparison with DINO-WM, including FIOC-DINO and FIOC-R3M.

**Disentanglement Quality** In SpritesWorld [77], we perform a linear probing analysis by training a linear regression layer on top of the learned representations to predict ground-truth static and dynamic factors. Static factors include object color and shape (encoded as one-hot vectors), while dynamic factors consist of object positions and velocities.

**Policy Learning** We consider both the policy performance on single-task and the generalization task. For generalization tasks, we consider three types of generalization: (1) *Attribute Generalization*: we evaluate for zero-shot generalization on new composition of object attributes; (2) *Object Attribute Composition*: we train models on domains with specific combinations of object attributes (e.g., color, shape, or material) and test them on domains with unseen attribute combinations; and *Skill Composition Generalization*: we train models on tasks with simple combinations of skills and test them on tasks requiring new combinations of skills. For all tasks, we use the average success rate as the evaluation metric.

## 5.2 Results on Learning World Models.

As evaluation of the learned dynamics and observations, Table 1 reports the LPIPS metric (Full Results are in Table A3). Compared to the baselines, our method achieves comparable or better reconstruction performance, particularly on the Fetch and Libero environments, where object interactions and dynamics are complex. Full results are in Appendix D.1.

We report also results on the accuracy of the learned interactions, quantified by the normalized Structured Hamming Distance (SHD) between the inferred interaction structures $\hat{\mathcal{G}} = \{\hat{G}_1, \ldots, \hat{G}_T\}$ and the ground truth structures. Fig. 6(a) presents the results on attribute and compositional generalization. For each bar, the shaded areas represent the performance of single-task learning with the same number of objects. The gap between the top of each bar and the top of the corresponding shaded area quantifies the performance drop when generalizing to novel scenarios (i.e., *empirical generalization gap*). These results demonstrate that FIOC consistently outperforms attention-based methods across all cases, verifying the importance of explicitly modeling the interaction structures and their changes using regime variables. And importantly, FIOC demonstrates superior generalization compared to attention-based methods, as shown by the smaller empirical generalization gap. Among the three versions of FIOC, all achieve strong attribute-level and compositional generalization. Notably, the variational masks with categorical distributions perform best, particularly in scenarios with a large number of objects. Full results are in Appendix D.1.

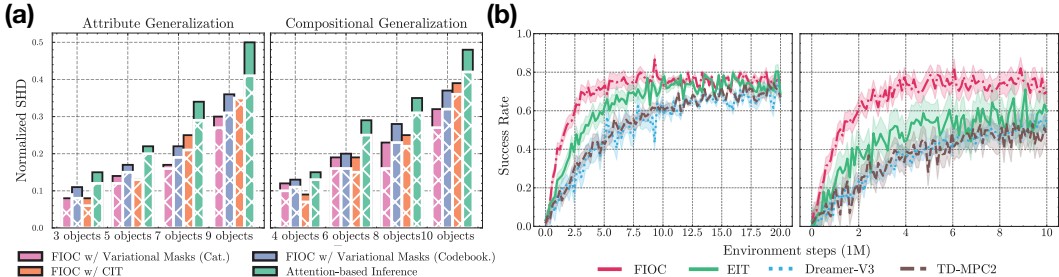

Figure 6: (a) Evaluation of learned interaction graphs with Normalized SHD for attribute and compositional generalization on Sprites-World with multiple objects. The shaded areas show results with the same number of objects for single-task learning. Lower values mean better performance. (b) Learning curves of single-task learning for i-Gibson (left) and Libero (right).

Fig. 5(a) reports the linear probing MSE of the learned static and dynamic representations, $c$ and $d$, against ground-truth attributes in the Sprites-world environment. We evaluate both the DINO-v2 raw input features and the object-centric DINO features (obtained from our first-stage learning without disentanglement). Our method achieves the best factorization of attributes: $c$ and $d$ effectively capture useful representations for dynamic features (i.e., position & velocity) and static features (i.e., color & shape), respectively. Notably, the object-centric DINO generally outperforms vanilla DINO on dynamic features. However, object-centric clustering tends to degrade static attribute representations such as color and shape. Our disentanglement module addresses this limitation by improving the representation of static attributes within each object.

|  | Envs | FIOC | Dreamer-V3 | EIT | TD-MPC2 |
|---|---|---|---|---|---|
| **Attri. Gen.** | Push & Switch | $0.91 \pm 0.05$ | $0.90 \pm 0.07$ | $\underline{0.92} \pm 0.04$ | $\mathbf{0.95} \pm 0.02$ |
|  | i-Gibson | $\mathbf{0.79} \pm 0.13$ | $0.62 \pm 0.16$ | $\underline{0.70} \pm 0.14$ | $0.65 \pm 0.15$ |
|  | Libero | $\mathbf{0.76} \pm 0.14$ | $0.59 \pm 0.18$ | $\underline{0.73} \pm 0.12$ | $0.69 \pm 0.18$ |
| **Comp. Gen.** | Push & Switch | $\mathbf{0.86} \pm 0.10$ | $0.81 \pm 0.12$ | $\underline{0.83} \pm 0.02$ | $0.79 \pm 0.08$ |
|  | Libero | $\mathbf{0.70} \pm 0.09$ | $0.58 \pm 0.12$ | $\underline{0.65} \pm 0.08$ | $0.63 \pm 0.14$ |
| **Skill Gen.** | Push & Switch | $\mathbf{0.81} \pm 0.06$ | $0.66 \pm 0.10$ | $\underline{0.73} \pm 0.08$ | $0.65 \pm 0.13$ |
|  | Franka Kitchen | $\mathbf{0.73} \pm 0.06$ | $0.59 \pm 0.09$ | $\underline{0.65} \pm 0.18$ | $0.62 \pm 0.08$ |

Table 2: Policy learning (success rate) of world model in Gym Fetch, Franka Kitchen, i-Gibson, and Libero tasks.

## 5.3 Results on Policy Learning.

Fig. 6(b) presents the learning curves (sampled every 100 time steps) on the i-Gibson and Libero tasks. The results indicate that world models incorporating object interactions, such as FIOC and EIT, achieve faster convergence compared to state-of-the-art methods like Dreamer-V3 and TD-MPC2. FIOC not only converges faster than EIT but also achieves a higher final success rate on Libero.

Fig.5(b) presents the offline RL results, comparing our method with DINO-WM [27], along with two variants of FIOC that use DINO-v2 [38] and R3M [39] as pre-trained visual embeddings. The results demonstrate that our approach achieves superior performance in both single-task learning and generalization, highlighting the advantages of the proposed two-level factorization on top of pre-trained visual features and the use of a hierarchical policy.

Table 2 presents the results of policy learning on single tasks, as well as those in the context of attribute, compositional, and skill generalization. The results indicate that FIOC performs comparably or better than other baselines in single-task learning scenarios and consistently outperforms them in all generalization tasks. Among the baselines, EIT achieves the second-best performance across generalization tasks. Detailed task settings are provided in Appendix F. The full results are in Table A2 and Fig. A3 in the appendix.

## 5.4 Ablation Studies

To evaluate the effectiveness of different components in both offline world model learning and online policy learning, we conduct a series of ablation studies on the following aspects. For the world model part, we consider cases: *Without state factorization:* The state $s$ is not factorized into static and dynamic components. Instead, the state transition $P_s$ is learned directly on the original $s$ obtained from the DINO embeddings. *Without interaction modeling:* Instead of modeling dynamic interactions, we assume a fully connected graph for all time steps and learn the dynamics using this dense graph; and *Using random actions in offline learning:* Instead of using pre-trained policies, we train the model with random actions

|  | Success Rate | |
| Ablations | Single Task | Comp. Gen. |
|---|:---:|:---:|
| FIOC | 0.81 | 0.70 |
| w/o Factorization | 0.77($\downarrow$ 0.04) | 0.64($\downarrow$ 0.06) |
| w/o Interaction | **0.63**($\downarrow$ **0.18**) | 0.52($\downarrow$ 0.18) |
| w/ random actions | 0.64($\downarrow$ 0.17) | **0.48**($\downarrow$ **0.22**) |
| w/o hierarchical policy | **0.58**($\downarrow$ **0.23**) | **0.42**($\downarrow$ **0.28**) |
| w/o pre-trained $\pi^l$ | 0.69($\downarrow$ 0.12) | 0.59($\downarrow$ 0.11) |
| w/o diversity | 0.62($\downarrow$ 0.19) | 0.50($\downarrow$ 0.20) |

Table 3: Ablation studies on Libero, evaluating the impact of removing specific components from the world model and policy learning. The **bold** entries indicate the ones with the largest performance drop. Light yellow and green areas represent the world model and policy learning components.

in the offline learning phase. For the policy learning stage, we consider the cases: *Without hierarchical policy:* Policy learning is performed directly on low-level actions without a high-level policy governing the sequence of interactions. *Without pre-trained low-level policy:* The low-level policy $\pi^l$ is not trained during the offline phase but learned from scratch in the online phase. *Without diversity term:* The diversity term in high-level policy learning is disabled.

The results in Table 3 show that for world models, interaction modeling and using the pre-trained policies in offline learning are the most critical components, as their removals lead to the most significant drop in policy learning performance. For policy learning, the hierarchical policy plays the most essential role. Other components, such as state factorization, utilizing pre-trained policies instead of random actions in the offline learning phase, pre-training the low-level policy, and incorporating diversity, also contribute to improving the policy learning performance.

## 6    Conclusions and Discussion

We study which types and degrees of decomposition make latent representations effective for efficient and generalizable policy learning. To this end, we introduce the Factored Interactive Object-Centric World Model (FIOC-WM), which learns a two-level factorization: an object-level representation with explicit interactions and an attribute-level representation for each object. FIOC-WM learns these decomposed structures directly from observations and leverages the resulting composable interaction primitives to enhance planning and policy learning via a hierarchical RL approach. The framework exhibits strong compositional generalization across attributes, objects, and skills, demonstrating that explicit, object-centric interaction decomposition is a key inductive bias for robust control.

**Limitations and Future Works**    FIOC-WM still relies on a pretrained object-centric model for object discovery, and its interaction models primarily generalize to seen object categories. Addressing these limitations and extending the framework to real-world robotic settings is part of the future work, potentially leveraging recent advances in robot-learning foundation models [88–97].

## Acknowledgment

We would like to thank the anonymous reviewers for their helpful comments and suggestions during the review process. We also acknowledge the computational support provided by the IVI servers at the University of Amsterdam and the University HPC centers at City University of Hong Kong.

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

# Appendix

## A Overview

In this appendix, we provide supplementary details, extended discussions, and full results for FIOC-WM. Specifically, Section B presents an in-depth discussion of related work, including factored and object-centric reinforcement learning, hierarchical RL, and causality-inspired RL, all of which are relevant to our approach. Section C offers a detailed analysis of world model learning, focusing on the two-level factorization of state transitions (Section C.1), interaction modeling (Section C.2), and policy learning (Section C.3). Sections D, E, and F cover the experimental results, network architectures, and task specifications.

## B Extended Related Works

### B.1 Factored and Object-centric Reinforcement Learning

Our paper takes inspiration from multiple factorization frameworks in RL. Factored RL is based on the model of a Factored Markov Decision Process (MDP) [59], where structural relationships exist

among states, actions, and rewards. This factorization enables efficient policy solutions by leveraging these structural relationships [60, 61].

A specific type of factorization, which we also adopt, is object-based, referred to as object-centric reinforcement learning (object-centric RL) [62], where states or observations are grouped into object-centric clusters. In object-centric RL, actions typically target only a subset of objects, and rewards are often associated with the states of specific objects or object subsets. This object-wise factorization facilitates more structured and efficient decision-making.

Recent works on object-centric RL can be broadly categorized into two major directions: (i) learning object-centric representation and (ii) modeling the object relations and policy architectures for compositional generalization. For the first line, approaches focus on using object-centric representation learning techniques [48, 63–65] to extract meaningful object-level features from raw observations. These methods, based on object-centric representations, focus on policy learning that leverages the object-centric encodings to learn object-centric policies either directly from object-centric representations [66, 98, 22, 99]. Haramati et al. [67] further Consider the object interactions and learn the policy that has compositional generalization with the model-free framework, learning directly from images. The second line of work develops object-centric policies modeling object relations and policy structures, incorporating inductive biases in the state transition and policy networks. Methods include the use of graph neural networks [68], linear relational networks [69], self-attention, and deep sets [70], and then they leverage the modeled object-centric states and relational structures to achieve compositional generalization in reinforcement learning [71, 18, 33, 22, 73, 22]. Haramati et al. [67] further leverages object interactions to learn a policy with compositional generalization, using a model-free framework that learns directly from images. Our work combines ideas from both directions, which learn state factorization from the object-centric level (object-based factorization) and state level (dynamics and static factorization), and model the interactions among objects to achieve compositional generalization. Different from [22, 100], which explores a more fine-grained object-centric and attribute-level factorization, our approach demonstrates that a simpler dynamic-static factorization of objects is already sufficient for effective world model and policy learning, striking a balance between minimality and expressiveness. Additionally, we go beyond object-centric policies by learning an interaction-centric policy that leverages the learned interaction model to facilitate long-horizon task learning.

## B.2 Hierarchical Reinforcement Learning

Our work, using low-level and high-level policy for decomposing complex tasks into interaction learning, is relevant to hierarchical reinforcement learning (HRL). HRL typically consists of a high-level policy (often referred to as the option framework [74], sub-skills, or sub-goals in the literature) and a low-level policy, enabling the efficient learning of complex RL tasks (see a recent survey [75]).

Within the scope of HRL, the most relevant to our work is the line of research that focuses on learning goal-conditioned hierarchical policies or hierarchical skill discovery. Within the scope of HRL, the closest to our work is the line of research that focuses on learning goal-conditioned hierarchical policies or hierarchical skill discovery. Zadaianchuk et al. [66] proposes the goal-conditioned hierarchical policies with learning object-based hierarchical goals. Hierarchical skill discovery focuses on decomposing complex tasks into object-wise or object-interaction-based components. For instance, Wang et al. [37] use conditional independence testing to identify sub-goals, while Chuck et al. [35] and [76] use Granger causality and causal models to uncover hierarchical structures. Our work shares similarities with [35, 37], particularly in using interactions as sub-skills. However, our work is on learning interaction models jointly with observations and dynamics within the world model directly from high-dimensional inputs, providing a more general and unified framework.

### B.2.1 Causality-inspired Reinforcement Learning

Closely related to factored RL, causality-based RL aims to learn and leverage causal structures in Markov Decision Processes (MDPs) [101]. Building causal structures within MDPs or world models can enable more efficient exploitation[102–105] or policy learning [54, 106]. Additionally, learned causal structures can facilitate counterfactual reasoning, providing benefits such as counterfactual data augmentation to improve RL efficiency [86, 87, 107, 108]. Similarly, our work builds an interactive world model that aligns with this line of research, utilizing factored and causal structures in dynamic

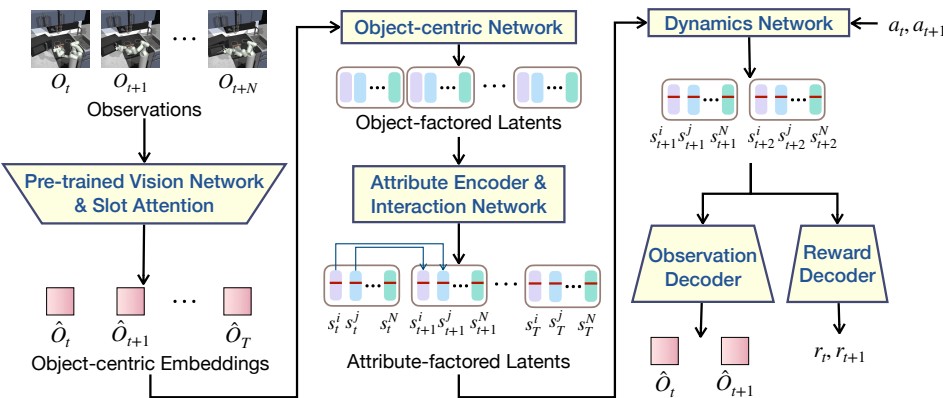

Figure A1: The pipeline of Offline Model Learning (Stage 1) jointly learns the observation function, state factorization, dynamics model, and reward model. Although Fig. 2 includes low-level policy learning as part of Stage 1, for clarity, we defer the discussion of low-level policy learning to Stage 2.

models to enhance generalization. Specifically, we focus on achieving compositional generalization at the levels of objects and their attributes.

The most relevant work to ours is SKILD [37], which also employs interaction-based hierarchical policies. However, there are several key differences. First, we aim to develop a general framework that incorporates state factorization, latent interaction-relevant states, and multiple approaches for learning interactions, including directly from pixel observations. In contrast, SKILD primarily focuses on state-based settings and learns interactions using conditional independence testing. Second, while SKILD is designed for unsupervised RL and skill discovery, our work focuses on general RL settings, although we consider extending it to unsupervised RL in future work. Despite these differences in scope and objectives, we acknowledge the contributions of SKILD, particularly in policy learning, and directly adopt certain components such as diversity measurement parameters and settings.

## C    Overall Framework

Fig. A1 illustrates the overall framework for learning FIOC-WN. In this section, we provide more supplementary details on this two-level factorization and interaction learning.

### C.1    Graphical Representation of State Transitions

Fig. A2 provides an illustrative example that complements the graphical model in Fig. 2 of the main paper, showing state transitions under dynamic graph structures (dashed red edges). We learn a two-level factorization of object-level and attribute-level representations, as detailed in Section 3.1. Here, we further motivate the choice of this two-level structure. First, decomposing a scene into individual objects reduces model complexity, as many objects move independently in most scenarios. By factoring dynamics and static attributes, the model can focus on learning the evolution of dynamic properties (e.g., position, velocity), while separately accounting for how static attributes (e.g., shape, mass) influence those dynamics. Second, to model interactions precisely and compactly, we focus on dynamic features being influenced by the attributes of interacting objects. For example, when two balls collide, it is primarily their dynamic attributes (e.g., velocity) that change, influenced by the full set of features (e.g., mass, shape, velocity) of the other object. This targeted interaction modeling allows the world model to be more precise and interpretable.

As a result, the learned world model benefits from this minimal yet sufficient factorization, which also enables more effective policy learning. The effectiveness of this design is further supported by ablation results shown in Table 3 in the main paper.

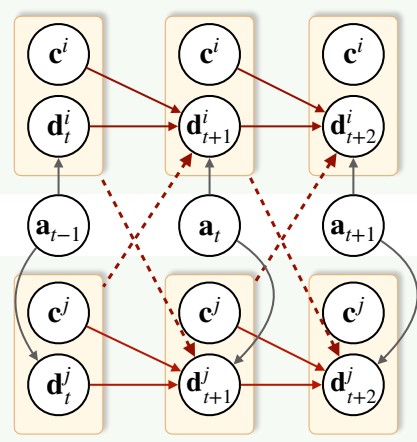

Figure A2: The detailed structure (example) of the state transitions with dynamic graph structure.

## C.2 Learning the Interaction Models

### C.2.1 Learning with Variational Masks

We consider two cases, where the masks are sampled from the approximated categorical distribution or from a codebook. In both cases, we have the ELBO [3]:

$$\mathcal{L}_{\text{mask}} = \mathbb{E}_{q_{\phi_u}(G|s)}\left[\log p_{\theta_u}(\mathbf{s} \mid G)\right] - \text{KL}\left[q_{\phi_u}(G \mid \mathbf{s})\|p(G)\right] \tag{A1}$$

The encoders are parameterized by graph neural networks, following the settings in neural relational inference [84, 85] and amortized causal discovery frameworks [52]. $P(G)$ is the prior distribution of the graph structures. Specifically, for the encoder $f_{\text{enc}}, \phi_u$, for each pair of nodes $i$ and $j$, we compute:

$$\mathbf{u}_{ij} = f_{\text{enc},\phi_u}(\mathbf{s}^i, \mathbf{s}^j). \tag{A2}$$

For the categorical distribution one, we have:

$$G^{i,j} \sim \text{Softmax}\left(\mathbf{u}_{ij} + g/\tau\right), \tag{A3}$$

where $\tau$ is the temperature parameter and $g$ is Gumbel-distributed noise [109].

For the latent codebook, we consider $u$ as the learned latent embedding, while maintaining a codebook prototype vector set $\mathbf{e} = \{\mathbf{e}_1, \ldots, \mathbf{e}_k\}$, where $k$ represents the number of different graph types (which can be interpreted as interaction patterns). Following [53], we apply a quantization step to discretize $u$:

$$e = e_z, \quad \text{where } z = \underset{j \in [k]}{\arg\min} \|\mathbf{u} - \mathbf{e}_j\|_2. \tag{A4}$$

After quantization, we retrieve the corresponding interaction graph by decoding the selected codebook entry into an adjacency matrix $G$:

$$G \sim g_{\text{dec}}(\mathbf{e}_z). \tag{A5}$$

All additional constraints and optimization techniques are directly adopted from [53].

---

[3]For clarity, in this section, we omit state factorization indices, as well as object and time indices, whenever they are not essential for computation or when they generally apply to factored variables across all objects and time steps.

### C.2.2 Learning with Conditional Independence Testing

Following [54], we use conditional independence testing to identify object interactions. Specifically, for each pair of objects $i$ and $j$, we compute the conditional mutual information (CMI) of the parameterized dynamics at each time step $t$:

$$\text{CMI}_{i,j} = \mathop{\mathbb{E}}_{\mathbf{s}_t, \mathbf{a}_t, \mathbf{s}_{t+1}^j} \left[ \log \frac{p_s \left( \mathbf{s}_{t+1}^j \mid \mathbf{a}_t, \mathbf{s}_t \right)}{p_s \left( \mathbf{s}_{t+1}^j \mid \left\{ \mathbf{a}_t, \mathbf{s}_t \setminus \mathbf{s}_t^i \right\} \right)} \right]$$

where $p_s$ represents the state transition dynamics used in our framework. During testing, we use the same parameterized transition model. Instead, after computing CMI, $G$ is encoded as the adjacent matrix in $\mathbb{R}^{N \times N}$, directly determining the interaction structure $G$. Since we employ an object-wise factorization, the number of required CMI tests remains manageable.

### C.3 Online Policy Learning

In policy learning tasks, we only use the variational masks for learning the regime variables.

**Low-level Policy** For the low-level policy that invokes interactions, we consider two approaches: model predictive control (MPC) and RL policies.

At time step $t$, we are given the target interaction graph at $t + k$, denoted as $G_t^k = G^*$. Given the learned transition model $P_u$, we use $s^i$ and $\mathbf{u}^g$ to infer the target states $s_g^i$ and $s_g^j$. Using these inferred target states, we apply MPC to generate a sequence of actions that transitions the system from $t$ to $t + k$ while minimizing the discrepancy between the predicted and target states.

Following Bharadhwaj et al. [110], Zhou et al. [27], we employ the cross-entropy method (CEM) for optimization. Specifically, we minimize the mean squared error (MSE) between the predicted state $\hat{\mathbf{s}}_{t+k}$ and the target state $\mathbf{s}_g$, given an action sequence $\mathbf{a}_t, \ldots, \mathbf{a}_{t+k-1}$ and the learned transition dynamics $p_s$:

$$\mathcal{L}_{\text{MPC}} = \|\mathbf{s}_{t+k} - \mathbf{s}_g\|_2^2. \tag{A6}$$

At each iteration, we sample a population of action sequences from a Gaussian distribution and use the MSE loss to update the mean and covariance of the distribution via stochastic gradient descent.

For RL policies, we train the policy $\pi^l(\mathbf{a}_t \mid \mathbf{s}_t, \mathbf{s}_g, \mathbf{u}_g)$ using PPO [57], similarly to the setting in [37].

**High-level Policy** For high-level policy, we use the diversity measurement, for each object $i$, we have this intrinsic motivation to sample the $j$ that has not been interacted. Hence, for an object set $\mathcal{S} = \{\mathbf{s}^1, \mathbf{s}^2, \ldots, \mathbf{s}^N\}$, at each time step $t$, we fix one object $s_i$ and sample a subset $\mathcal{S}_t \subseteq \mathcal{S}$ based on the diversity reward $r_{\text{div}}$ introduced that prioritizes diversity and avoids already-interacted objects. We learn this policy via PPO, using the same way as those in SKILD [106] but with the additional task reward.

## D Full Results

### D.1 Full Results on World Modeling

**Interaction Learning** Table A1 gives the full results on the interaction learning. We use Structural Hamming Distance (SHD) to verify the effectiveness of capturing interactions. SHD is a standard metric widely adopted in the relational inference and causal discovery literature [111, 112, 53]. Results show that the variational method with a categorical distribution as the prior achieves the best performance, while the variational approach with codebook latents performs second-best across domains. Conditional independence testing (CIT) also yields comparable results to these two methods. In contrast, relying solely on attention-based mechanisms is not robust across all settings, indicating that directly inferring interactions from the attention matrix is insufficient. For precise interaction modeling, the relational inference modules are necessary.

**Algorithm 1** FIOC-WM: Offline World Model and Online Policy Learning (Simplified)

---

**Require:** Offline dataset $\mathcal{D} = \{(o_t, a_t, r_t)\}$; pre-trained visual encoder $p_{\text{pre}}$; inference model $q_\phi$; transition model $p_s$; reward model $p_r$; interaction model $p_u$; high-level policy $\pi^h$; interaction (low-level) policy $\pi^\ell$

**Ensure:** Policies $\pi^h, \pi^\ell$ and world model components

1: **Stage 1: Offline World-Model Learning**
2: **for all** $(o_t, a_t, r_t) \in \mathcal{D}$ **do**
3:     $\hat{o}_t \leftarrow p_{\text{pre}}(o_t)$                          ▷ Encode observation with pre-trained vision model
4:     $s_t^i \leftarrow q_\phi(s_t^i \mid \hat{o}_t)$                        ▷ Infer object-centric latent state
5:     Factorize $s_t^i$ into $s_t^i = (d_t^i, c^i)$               ▷ dynamics- and attribute-level factors
6:     Learn reward $p_r(r_t \mid s_t, a_t)$
7:     Learn transition $p_s(s_{t+1} \mid s_t, a_t, G_t)$
8:     Infer interaction graphs $G_t \sim p_u(G_t \mid s_t)$
9:     Train interaction policy $\pi^\ell(a_t \mid s_t, G_t^g)$
10: **end for**

11: **Stage 2: Online Policy Learning**
12: **repeat**                                                    ▷ For each environment rollout episode
13:     Observe environment steps; encode current state $s_t$ using world model
14:     $G_t^g \sim \pi^h(G_t^g \mid s_t)$                      ▷ Select goal/target interaction graph (object-centric subgoal)
15:     $a_t \sim \pi^\ell(a_t \mid s_t, G_t^g)$                    ▷ Sample low-level action conditioned on interaction
16:     Execute $a_t$; observe $r_t, o_{t+1}$; update $s_{t+1}$
17:     Update policies $\pi^h, \pi^\ell$ with collected data
18: **until** episode terminates

---

|  | Envs | Variational (Cat.) | Variational (Code.) | CIT | Attention-based |
|---|---|---|---|---|---|
| **Single-Task** | 3 objects | $0.09_{\pm 0.04}$ | $\underline{0.08}_{\pm 0.03}$ | $\mathbf{0.06}_{\pm \mathbf{0.02}}$ | $0.12_{\pm 0.04}$ |
|  | 5 objects | $\mathbf{0.12}_{\pm \mathbf{0.07}}$ | $0.15_{\pm 0.09}$ | $\underline{0.13}_{\pm 0.06}$ | $0.20_{\pm 0.07}$ |
|  | 7 objects | $\mathbf{0.16}_{\pm \mathbf{0.10}}$ | $\underline{0.19}_{\pm 0.08}$ | $0.21_{\pm 0.07}$ | $0.29_{\pm 0.12}$ |
|  | 9 objects | $\mathbf{0.27}_{\pm \mathbf{0.10}}$ | $\underline{0.31}_{\pm 0.09}$ | $0.35_{\pm 0.15}$ | $0.41_{\pm 0.14}$ |
| **Attri. Gen.** | 3 objects | $\mathbf{0.08}_{\pm \mathbf{0.02}}$ | $0.11_{\pm 0.05}$ | $\underline{0.08}_{\pm 0.03}$ | $0.15_{\pm 0.02}$ |
|  | 5 objects | $\underline{0.14}_{\pm 0.06}$ | $0.17_{\pm 0.11}$ | $\mathbf{0.13}_{\pm \mathbf{0.04}}$ | $0.22_{\pm 0.11}$ |
|  | 7 objects | $\mathbf{0.17}_{\pm \mathbf{0.12}}$ | $\underline{0.22}_{\pm 0.13}$ | $0.25_{\pm 0.06}$ | $0.34_{\pm 0.15}$ |
|  | 9 objects | $\mathbf{0.30}_{\pm \mathbf{0.10}}$ | $0.36_{\pm 0.14}$ | $\underline{0.35}_{\pm 0.19}$ | $0.50_{\pm 0.18}$ |
| **Comp. Gen.** | 4 objects | $\underline{0.12}_{\pm 0.06}$ | $0.13_{\pm 0.04}$ | $\mathbf{0.09}_{\pm \mathbf{0.03}}$ | $0.15_{\pm 0.06}$ |
|  | 6 objects | $\underline{0.19}_{\pm 0.09}$ | $0.20_{\pm 0.10}$ | $\mathbf{0.19}_{\pm \mathbf{0.07}}$ | $0.29_{\pm 0.11}$ |
|  | 8 objects | $\mathbf{0.23}_{\pm \mathbf{0.13}}$ | $0.28_{\pm 0.12}$ | $\underline{0.5}_{\pm 0.09}$ | $0.35_{\pm 0.14}$ |
|  | 10 objects | $\mathbf{0.32}_{\pm \mathbf{0.14}}$ | $\underline{0.37}_{\pm 0.12}$ | $0.39_{\pm 0.13}$ | $0.48_{\pm 0.16}$ |

Table A1: Full results on the normalized SHD of FIOC with different interaction learning models in predicting the ground truth interactions in Sprites World Environment. The **bold** values indicate the best-performing method, and the underlined ones are the second-best.

**Reconstruction** Table A3 presents the complete LPIPS results across domains. LPIPS [83] (lower is better) evaluates the comparison between the generated frames with ground-truth observations at both the pixel and perceptual levels. Our FIOC model achieves the best performance in most cases, particularly in environments with rich interactions such as Sprites, Fetch, and Libero. In other cases where DINO-WM performs best, our model remains competitive, with LPIPS scores within 0.05 of DINO-WM.

### D.2   Policy Learning

Table A2 gives the full results on single-task learning, attribute generalization, compositional generalization, and skill generalization in policy learning tasks. The learning curves of single task learning are given in Fig. A3. For some simpler tasks in these single-task learning scenarios, existing baselines, particularly EIT and TD-MPC2, can achieve strong performance, for example, on Fetch. However,

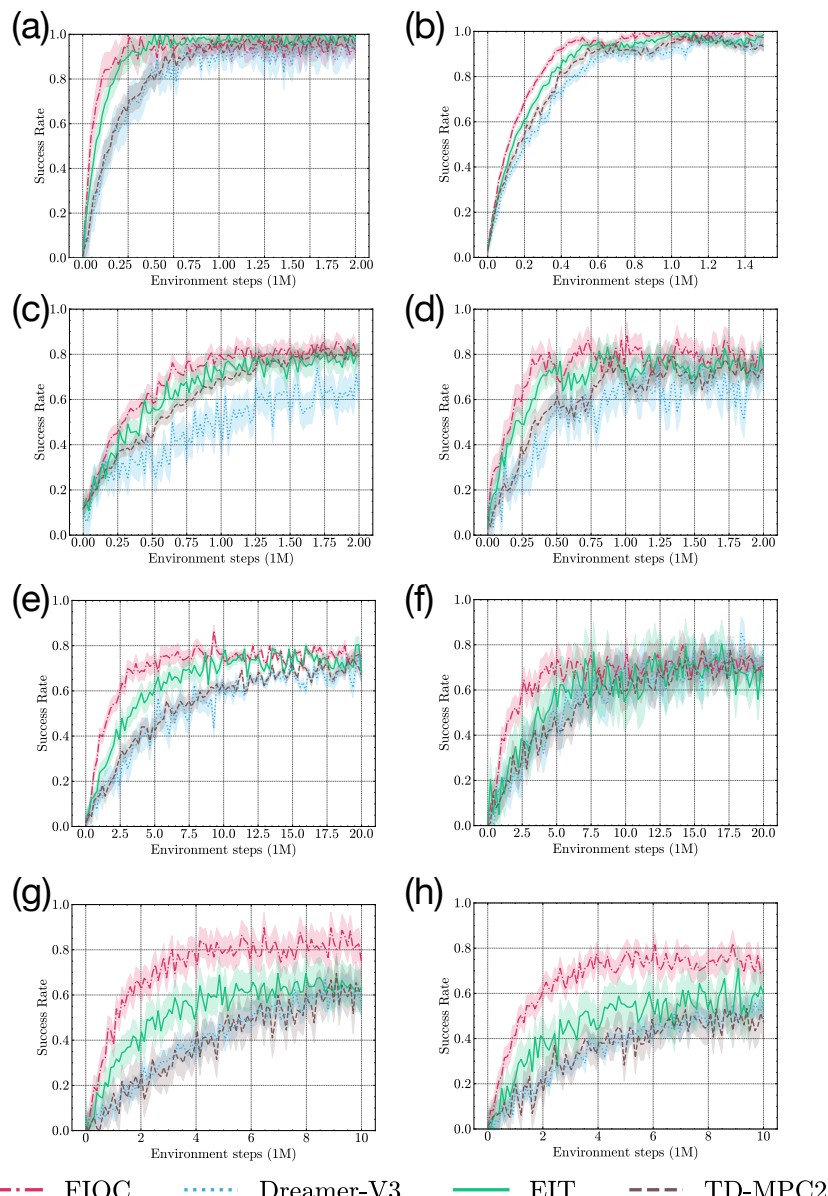

Figure A3: The policy learning curves of single task learning scenarios, (a). Gym Fetch Task 1; (b) Gym Fetch Task 2; (c) Franka Kitchen Task 1; (d) Franka Kitchen Task 2; (e). i-Gibson Task 1; (f). i-Gibson Task 2; (g). Libero Task 1; (h). Libero Task 2.

our method achieves the best performance in 5 out of 8 tasks, second-best in 1 task, and comparable performance (all $> 90\%$ success rate) on the remaining 2 tasks, which are relatively simple.

### D.3 Full Ablations

**Online World Model Fine-tuning**   For computational efficiency and because our experiments indicate that an offline-trained world model is already robust given high-quality offline data, we do not update the world model during the online stage by default. This is an empirical choice rather than a limitation; online updates are feasible. To assess the impact, we perform an ablation comparing performance with and without online world-model updates. Results are shown in Table A4.

| | Envs | FIOC | Dreamer-V3 | EIT | TD-MPC2 |
|---|---|---|---|---|---|
| **Single-Task** | Gym Fetch (Task 1) | $0.95_{\pm0.03}$ | $\mathbf{0.98_{\pm0.02}}$ | $0.93_{\pm0.05}$ | $\underline{0.97}_{\pm0.02}$ |
| | Gym Fetch (Task 2) | $0.98_{\pm0.01}$ | $\mathbf{0.97_{\pm0.02}}$ | $0.95_{\pm0.02}$ | $\underline{0.96}_{\pm0.02}$ |
| | Franka Kitchen (Task 1) | $\underline{0.82}_{\pm0.04}$ | $0.75_{\pm0.06}$ | $0.69_{\pm0.07}$ | $\mathbf{0.83_{\pm0.03}}$ |
| | Franka Kitchen (Task 2) | $\underline{0.79}_{\pm0.06}$ | $0.68_{\pm0.09}$ | $0.75_{\pm0.08}$ | $\mathbf{0.73_{\pm0.04}}$ |
| | i-Gibson (Task 1) | $\mathbf{0.76_{\pm0.12}}$ | $0.69_{\pm0.19}$ | $\underline{0.74}_{\pm0.14}$ | $0.72_{\pm0.12}$ |
| | i-Gibson (Task 2) | $\mathbf{0.72_{\pm0.10}}$ | $0.68_{\pm0.19}$ | $\underline{0.74}_{\pm0.14}$ | $0.72_{\pm0.12}$ |
| | Libero (Task 1) | $\mathbf{0.81_{\pm0.11}}$ | $0.65_{\pm0.14}$ | $\underline{0.78}_{\pm0.09}$ | $0.76_{\pm0.14}$ |
| | Libero (Task 2) | $\mathbf{0.74_{\pm0.09}}$ | $0.58_{\pm0.16}$ | $\underline{0.69}_{\pm0.07}$ | $0.65_{\pm0.12}$ |
| **Attri. Gen.** | Push & Switch | $0.91_{\pm0.05}$ | $0.90_{\pm0.07}$ | $\underline{0.92}_{\pm0.04}$ | $\mathbf{0.95_{\pm0.02}}$ |
| | i-Gibson | $\mathbf{0.79_{\pm0.13}}$ | $0.62_{\pm0.16}$ | $\underline{0.70}_{\pm0.14}$ | $0.65_{\pm0.15}$ |
| | Libero | $\mathbf{0.76_{\pm0.14}}$ | $0.59_{\pm0.18}$ | $\underline{0.73}_{\pm0.12}$ | $0.69_{\pm0.18}$ |
| **Comp. Gen.** | Push & Switch | $\mathbf{0.86_{\pm0.10}}$ | $0.81_{\pm0.12}$ | $\underline{0.83}_{\pm0.02}$ | $0.79_{\pm0.08}$ |
| | Libero | $\mathbf{0.70_{\pm0.09}}$ | $0.58_{\pm0.12}$ | $\underline{0.65}_{\pm0.08}$ | $0.63_{\pm0.14}$ |
| **Skill Gen.** | Push & Switch | $\mathbf{0.81_{\pm0.06}}$ | $0.66_{\pm0.10}$ | $\underline{0.73}_{\pm0.08}$ | $0.65_{\pm0.13}$ |
| | Franka Kitchen | $\mathbf{0.73_{\pm0.06}}$ | $0.59_{\pm0.09}$ | $\underline{0.65}_{\pm0.18}$ | $0.62_{\pm0.08}$ |

Table A2: Policy learning (success rate) of world model in Gym Fetch, Franka Kitchen, i-Gibson, and Libero tasks.

| Environment | Dreamer-V3 | TD-MPC2 | EIT | DINO-WM | FIOC |
|---|---|---|---|---|---|
| Sprites | 0.026 | 0.019 | 0.006 | 0.012 | **0.004** |
| Fetch | 0.042 | 0.039 | 0.026 | 0.009 | **0.007** |
| Kitchen | 0.102 | 0.123 | 0.096 | **0.035** | 0.038 |
| i-Gbison | 0.135 | 0.092 | 0.085 | **0.063** | 0.068 |
| Libero | 0.089 | 0.061 | 0.040 | 0.035 | **0.027** |

Table A3: Comparison of world models on LPIPS metrics.

**Using pre-trained embeddings for Dreamer and TD-MPC**    Specifically, we adapted both baselines to use DINO embeddings as input:

- For Dreamer-V3, we replaced the pixel encoder with a frozen DINO encoder and used DINO-WM's decoder to reconstruct the observations.

- For TD-MPC2, we used DINO features to predict actions, terminal values, and rewards via its original decoding heads.

The updated results are shown in Table A5. The results indicate that the pre-trained DINO features improves both Dreamer-V3 and TD-MPC2 performance in some cases, but they still under-perform compared to FIOC. This suggests that while strong visual representations help, our proposed factorization (Stage 1) and policy learning (Stage 2) are key contributors to the performance gain.

**Generalize to more objects**    To assess the model's ability to generalize to a greater number of objects, we train FIOC on environments containing three objects and evaluate on tasks with six and eight objects while keeping the world model fixed in Fetch Env. As shown in Table A6, FIOC achieves strong generalization performance, comparable to or better than baselines such as EIT, and it consistently outperforms Dreamer-V3 and TD-MPC2 under distribution shifts in object count.

**Visualization Rollouts**    Some visualization rollouts are found at the project homepage: `https://sites.google.com/view/fioc-wm`.

| Envs | FIOC (w/o Online Tuning) | FIOC (w/ Online Tuning) |
|------|------|------|
| Gym Fetch (Task 1) | $0.95 \pm 0.03$ | $0.96 \pm 0.02$ |
| Gym Fetch (Task 2) | $0.98 \pm 0.01$ | $0.98 \pm 0.01$ |
| Franka Kitchen (Task 1) | $0.82 \pm 0.04$ | $0.79 \pm 0.06$ |
| Franka Kitchen (Task 2) | $0.79 \pm 0.06$ | $0.82 \pm 0.05$ |
| i-Gibson (Task 1) | $0.76 \pm 0.12$ | $0.78 \pm 0.14$ |
| i-Gibson (Task 2) | $0.72 \pm 0.10$ | $0.75 \pm 0.06$ |
| Libero (Task 1) | $0.81 \pm 0.11$ | $0.83 \pm 0.08$ |
| Libero (Task 2) | $0.74 \pm 0.09$ | $0.71 \pm 0.06$ |
| Push & Switch (Attri. Gen.) | $0.91 \pm 0.05$ | $0.95 \pm 0.08$ |
| i-Gibson (Attri. Gen.) | $0.79 \pm 0.13$ | $0.81 \pm 0.15$ |
| Libero (Attri. Gen.) | $0.76 \pm 0.14$ | $0.68 \pm 0.09$ |
| Push & Switch (Comp. Gen.) | $0.86 \pm 0.10$ | $0.82 \pm 0.12$ |
| Libero (Comp. Gen.) | $0.70 \pm 0.09$ | $0.74 \pm 0.08$ |
| Push & Switch (Skill Gen.) | $0.81 \pm 0.06$ | $0.82 \pm 0.08$ |
| Franka Kitchen (Skill Gen.) | $0.73 \pm 0.06$ | $0.72 \pm 0.07$ |

Table A4: Performance comparison of FIOC with and without online world model tuning across different environments. Values are mean $\pm$ standard deviation.

| Envs | FIOC | Dreamer-V3 (DINO/original) | TD-MPC2 (DINO/original) |
|------|------|------|------|
| Kitchen | 0.82 | 0.77 / 0.75 | 0.79 / 0.83 |
| i-Gibson | 0.76 | 0.71 / 0.69 | 0.73 / 0.72 |
| Libero | 0.81 | 0.69 / 0.65 | 0.74 / 0.76 |

Table A5: Comparison of FIOC with baselines using DINO features as input. Values indicate task success rates.

# E  Network Architectures and Hyper-parameters

## E.1  World Models

**Learning the Observation Functions**    For DINO-v2, we use ViT-Base for all cases. For R3M, we use ResNet-50 as backbones for all cases. For slot attention parameters, all settings remain consistent. Following VideoSAUR [47], we transform the original features using a two-layer MLP with an output dimension equal to the slot dimension. The slot attention module is initialized with randomly sampled slots to group the first-frame features. For subsequent frames, we initialize the slot attention module with the slots from the previous frame, which are additionally transformed using a predictor module with a GRU recurrent unit in the slot attention grouping.

For the VAE used to learn latent states, we employ a two-layer MLP with a hidden size of 256. The specific hyperparameters for different environments are detailed in Table A7.

**Learning the Regime Variables**    For variational masks with a categorical distribution, we directly adopt the hyperparameters and network design from ACD [52]. However, unlike ACD, we use a GRU as the encoder, where the MLP has a hidden size of 256 with 3 layers. For the codebook-based approach, we use an MLP with 3 layers. The hidden layer size is set to 128 for Sprites-World, while for other environments, it is 256. The number of the centered codes are 16 for Sprites World, 8 for Gym Fetch, and 10 for others. All training hyperparameters follow those specified in [52] and [53].

For conditional independence testing, the threshold hyperparameters are set as follows: 0.02 for Sprites-World, 0.15 for Gym-Fetch, and 0.05 for other environments. All remaining hyperparameters are shared across environments.

**Learning the State Transitions**    The state transitions are modeled using MLP layers with different configurations across environments. Specifically, we use a 2-layer MLP with hidden dimensionality 32 and SiLU activation for Sprites-World, a 2-layer MLP with hidden dimensionality 64 for Gym-

| Envs | FIOC (Ours) | Dreamer-V3 | EIT | TD-MPC2 |
|------|-------------|------------|-----|---------|
| 3 objects | 0.93 | 0.96 | 0.94 | **0.97** |
| 6 objects | **0.81** | 0.54 | 0.77 | 0.62 |
| 8 objects | **0.70** | 0.44 | 0.62 | 0.53 |

Table A6: Generalization to increased object count. Models are trained with 3 objects and evaluated with 6 and 8 objects using a fixed world model. FIOC shows strong generalization, matching or exceeding EIT and outperforming Dreamer-V3 and TD-MPC2 under distribution shifts in object count.

| Parameter | Values (shared if not specified) |
|-----------|----------------------------------|
| Used VIT for DINO | Base |
| Used ResNet for R3M | ResNet-50 |
| Patch Size | 16 |
| Feature Dimension | 768 |
| Gradient Norm Clip | 0.05 |
| Image Crop/Resize | 64 (SpritesWorld), 224 (others) |
| Slots | Number of objects + 2 |
| Iterations for Clustering | 3 |
| Slot Dimension | 32 (SpritesWorld), 64 (Gym-Fetch), 128 (others) |
| Latent Dimensions for $s^s, s^c$ | 8, 6 (SpritesWorld); 10, 8 (Gym-Fetch); 16, 12 (others) |

Table A7: Hyperparameters used in learning observation functions.

Fetch, and a 3-layer MLP with hidden dimensionality 128 for other environments. The one-step prediction GRU layer consists of 3 MLP layers, with hidden dimensionality 128 for Sprites-World and 256 for i-Gibson, Gym-Fetch, Libero, and Franka Kitchen. The detailed settings are provided in Table A8. All with the learning rate $3e - 4$.

| Environment | MLP Layers (State Transition) | GRU Layers (One-Step Prediction) |
|-------------|-------------------------------|----------------------------------|
| Sprites-World | 2 layers, 32 hidden | 3 layers, 128 hidden |
| Gym-Fetch | 2 layers, 64 hidden | 3 layers, 256 hidden |
| i-Gibson | 3 layers, 128 hidden | 3 layers, 256 hidden |
| Libero | 3 layers, 128 hidden | 3 layers, 256 hidden |
| Franka Kitchen | 3 layers, 128 hidden | 3 layers, 256 hidden |

Table A8: Hyperparameters for state transition modeling across different environments.

**Offline RL** For offline RL experiments, we use the same hyper-parameters as the online ones. Same as DINO-WM [27], we do not use expert demonstrations and inverse dynamics models to learn the mapping $p(\mathbf{a}_t|\mathbf{s}_t, \mathbf{s}_{t+1})$.

**Others** For offline training, we collect 3000 episodes with random actions for Sprites-World. For all other environments, we collect 2000 episodes using pre-trained policies from Dreamer-v3. The hyperparameters for the loss terms are set as $\{\alpha, \beta, \gamma, \eta\} = \{1, 0.05, 0.1, 0.2\}$, and the learning rate is set to $3 \times 10^{-4}$. The detailed data collection settings are provided in Table A9.

### E.2 Policy Learning

**Low-Level Policy** For MPC, we use gradient descent with a learning rate of $5 \times 10^{-5}$. For those using PPO, we set the learning rate to $3 \times 10^{-4}$ with a clip ratio of 0.1. The MLP architecture consists of hidden sizes $[256, 256]$ for Gym-Fetch, while for other environments, we use $[512, 512]$. Generalized Advantage Estimation (GAE) is set to 0.95 for all environments, and the entropy coefficient is 0.1.

| Environment | Number of Episodes | Action Strategy |
|---|---|---|
| Sprites-World | 3000 | Random Actions |
| Gym-Fetch, i-Gibson, Libero, Franka Kitchen | 2000 | Pre-trained Policies (Dreamer-v3) |

Table A9: Offline training settings across different environments.

**High-Level Policy**  For high-level policy learning, we use PPO with a learning rate of $1 \times 10^{-4}$. The MLP architecture follows the same structure as the low-level policy, with hidden sizes of $[256, 256]$ for Gym-Fetch and $[512, 512]$ for other environments. The batch size is 1024 for all.

### E.3   Computes and Training Time

Compute used for training the FIOC-WM:

- For Sprites-World, we use 3 hours on 1x NVIDIA A100;
- For Fetch, we use 8 hours on 6x NVIDIA 4090;
- For i-Gibson, we use 9 hours on 6x NVIDIA 4090;
- For Libero-object, we use 8 hours on 1x NVIDIA A100;
- For Kitchen, we use 6 hours on 1x NVIDIA A100.

## F   Task Details

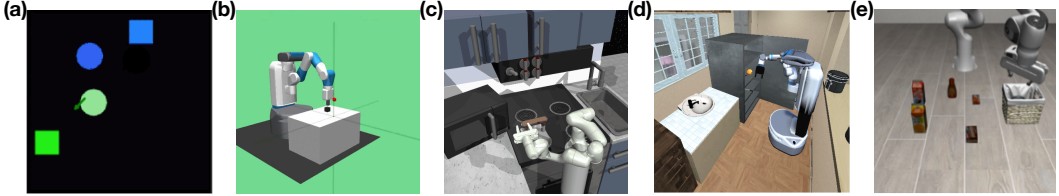

Figure A4: visualization of used benchmarks. From left to right: (a). Sprites-World; (b). OpenAI-Gym Fetch; (c). Franka Kitchen; (d). i-Gibson; and (e). Libero.

**OpenAI Gym Fetch** [78] is an environment featuring a Fetch robotic arm capable of manipulating cubes and switches. The tasks involve completing sub-tasks that require pushing or switching a varying number of objects. For single-task learning, we consider 2-push (Task 1) and 2-switch (Task 2), each with 2 million and 1.5 million training steps. For the attribute generalization task, we consider changing the color of the objects. For the compositional generalization task, we add one object to the push task, making it becoming the 3-push task. For skill one, we consider training both 2-push and 2-switch and compose them together for 2-push + 3-switch. All generalization tasks are evaluated with zero-shot generalization (for the skill generalization, we let the agent know the compositional task structure by providing separate rewards).

**Franka-kitchen** [40] is the environment where the 7-DoF Franka Emika Panda arm needs to perform tasks in the kitchen setup. Here we consider several sequential sub-tasks, such as turning on the microwave, moving the kettle, turning on the stove, and turning on the light. For single-task learning, we consider these two tasks:

- Task-1 is *Turn on the microwave - Move the kettle - Turn on the stove - Turn on the light*;
- Task-2 is *Turn on the microwave - Turn on the light - Slide the cabinet to the right - Open the cabinet*.

All tasks are with 2M training steps. The skill generalization one is *Turn on the microwave - Move the kettle - Slide the cabinet to the right - Open the cabinet*, evaluating with 0.2M training ($10\%$ as the base tasks).

**i-Gibson** [82] is a realistic environment with a simulated Fetch robot operating in everyday household tasks with rich objects and interactions. Similar to the setting in [37], we consider the tasks that

related to the peach object. The peach can be washed or cut, adding complexity to the tasks. The Task-1 is grasping the peach, Task-2 is cutting it with a knife. Each is with 20M steps to train. For attribute generalization, we change both the color and the size of the peach and follow the Task-1 setting with 1M adaptation steps.

**Libero** [80] is a benchmark designed for lifelong robot learning and imitation learning in household and tabletop environments. We focus on tasks randomly selected from the task library within libero-object. Task 1 and Task 2 involve picking two different sets of daily objects (boxes, cubes, and glasses) and moving them to a designated basket. The compositional generalization setting introduces objects with different colors and requires picking another randomly selected set of objects and placing them in the basket. The number of objects to manipulate is 5 for Task 1 and Task 2, while the generalization setting includes 7 objects. Number of training steps are 10M for the base tasks and 1M for the generalization task.

