# OpenReview forum: "Learning Interactive World Model for Object-Centric Reinforcement Learning"
_NeurIPS.cc/2025/Conference — NeurIPS 2025 poster_

### Official Review · Reviewer_gBUX · 2025-07-01

**Clarity:** 2
**Significance:** 3
**Originality:** 3
**Rating:** 4
**Confidence:** 3

**Summary:**

This paper proposes FIOC-WM (Factored Interactive Object-Centric World Model), a framework for learning structured object-based representations with explicit interaction modeling in reinforcement learning. FIOC-WM decomposes the environment into object-level interactions (e.g., collisions, containment) and attribute-level states (static vs. dynamic properties). It leverages pre-trained vision models and slot attention to extract object-centric representations from pixels, then uses a hierarchical policy for long-horizon reinforcement learning.

**Questions:**

1. The paper leverages pre-trained vision models like DINO-v2 and R3M to extract high-quality features. A clarification would be helpful as to whether these pre-trained models are fine-tuned during the world model learning stage.
2. The reconstruction loss is applied in the object-centric embedding space rather than the raw image space. The reasoning behind this choice could be further explained. While this approach is likely computationally efficient, it appears to heavily rely on the quality and generalization of the pre-trained model's features, which might lead to suboptimal learned representations if the pre-trained model's features are not sufficiently expressive for a given complex task.

**Ethical Concerns:**

["NO or VERY MINOR ethics concerns only"]

**Final Justification:**

Although the original version had poor readability, the authors provided a detailed explanation in their rebuttal and stated they will make changes in the camera-ready version. I am willing to increase my score by one point.

**Limitations:**

The authors addressed the limitations and potential negative societal impact of their work.

**Paper Formatting Concerns:**

There is no formatting issue.

**Quality:**

2

**Strengths And Weaknesses:**

Strengths:

The work proposes a two-level factorization of representations: an object-level factorization with interactions and an attribute-level factorization for each object (dynamic and static). This approach provides a principled inductive bias that reduces redundancy and highlights the minimal sufficient components needed for planning and control.


Weaknesses:

1. The paper is rich in technical detail, but the overall readability could be enhanced.
2. To further strengthen the paper's claims, it would be valuable to include some qualitative visualizations of the disentanglement effect, which could make the results more intuitive and compelling.
3. Figure 1 suggests that a low-level interaction policy is learned during the offline model learning stage. However, the description in Section 3.1 focuses on the world model's components and losses, and the discussion of this policy's learning is deferred to Stage 2. A clarification on whether the low-level policy is indeed trained offline.
4. In Section 3.2, it is stated that the policy is learned "by leveraging the predicted interactions to learn the inverse mapping from interactions to actions". A more detailed explanation of this inverse mapping process would be beneficial.
5. The use of Slot Attention is a critical component for learning object-centric representations. Given that the number of objects can vary across different environments, it would be helpful to know how the number of slots is determined. Conducting ablation studies on the effect of the number of slots on the model's performance in different environments would be better.

---

> ### Author Rebuttal · Authors · 2025-07-31
>
> Thank you for your constructive feedback and review! We truly appreciate your recognition of our methodology. Most of the concerns relate to technical details and clarity. For clarity-related issues, we provide concrete and executable plans that can be incorporated into the camera-ready version. For the technical concerns, we offer detailed clarifications along with additional evaluations in this rebuttal. Please find our responses below.
>
> ---
>
> > The paper is rich in technical detail, but the overall readability could be enhanced.
>
> **R1**: Thank you for the constructive feedback. To improve readability, we have devised a concrete revision plan for the camera-ready version, addressing each question below. Our focus is on clarifying the technical content and improving the overall flow, as suggested. Please refer to the tags “[**For clarity**]” in our responses to see where these revisions are proposed.
>
> ---
>
> >It would be valuable to include some qualitative visualizations of the disentanglement effect, which could make the results more intuitive and compelling.
>
> **R2**: Regarding the qualitative visualization of the disentanglement effect, we agree that the illustrations could be useful. We cannot provide it here as we are not allowed to add additional figures during rebuttal. In the camera-ready version, we will make sure to include visualizations of the reconstructed images to provide a more direct qualitative assessment. These will be included in the appendix.
>
> We would also like to emphasize that in the current submission, we primarily evaluated disentanglement through probing results (see Fig. 5(a)), as our goal is to learn disentangled representations directly from pre-trained models such as DINO and R3M, indicating that our method can extract structured object-centric information without additional supervision. In addition, we also reported LPIPS scores (Table 1), which reflect pixel-level reconstruction quality.
>
> ---
>
> > Figure 1 suggests that a low-level interaction policy is learned during the offline model learning stage. However, the description in Section 3.1 focuses on the world model's components and losses, and the discussion of this policy's learning is deferred to Stage 2. A clarification on whether the low-level policy is indeed trained offline.
>
> **R3**: The low-level policy $\pi^l$ is trained during the offline stage and then fine-tuned in the online stage. We acknowledge that this setup may have caused some confusion, as the details for learning $\pi^l$ are presented in Section 3.2.
>
> Our original intention was to clearly separate the components: first, explaining how the world model (including dynamics, observation, and reward) is learned, and then detailing how the hierarchical policy is constructed, in order to make the framework easier to follow.
>
> [**For clarity**] In the revision, we will address this potential misunderstanding by adding a clarifying sentence after Line 166: “Note that the learning of $\pi^l$ is performed in the offline stage and subsequently fine-tuned during the online stage.”
>
> And we plan to add the algorithm to the appendix (here is the brief sketch):
>
> **Offline World Model and Online Policy Learning Framework**
>
> ---
>
> **Stage 1: Offline Model Learning**
> For each $(o_t, a_t, r_t) \in \mathcal{D}$:
> - Encode observation: $\hat{o}_t \leftarrow q_\phi(o_t)$
> - Infer latent state: $s_t^i \leftarrow q_\phi(s_t^i \mid \hat{o}_t)$
> - Factorize latent state: $s_t^i = (d_t^i, c_t^i)$
> - Learn reward model: $p_r(r_t \mid s_t, a_t)$
> - Learn transition model: $p_s(s_{t+1} \mid s_t, a_t, G_t)$
> - Infer interaction graphs: $p_u(G_t \mid s_t)$
> - Train interaction policy: $\pi^l(a_t \mid s_t, G_t^g)$
>
> ---
>
> **Stage 2: Online Policy Learning**
> For each environment rollout episode:
> - Receive current state $s_t$ from world model encoder with input environmental steps
> - Select goal graph: $G_t^g \sim \pi^h(G_t^g \mid s_t)$
> - Sample action from interaction policy: $a_t \sim \pi^l(a_t \mid s_t, G_t^g)$
> - Execute $a_t$ in environment, observe $r_t$, $s_{t+1}$
> - Update policies $\pi^h$, $\pi^l$ using collected data
>
> ---
>
> > In Section 3.2, it is stated that the policy is learned "by leveraging the predicted interactions to learn the inverse mapping from interactions to actions". A more detailed explanation of this inverse mapping process would be beneficial.
>
> **R4**: The inverse mapping is realized through either MPC or PPO, as described later in Section 3.2 and Appendix C.3. Given a predicted target interaction graph at time $t+k$ (e.g., $G_t^k = G^*$), we treat it as the goal with state *s_g* and use the learned transition model to infer the target states. For MPC, we optimize the action sequence to minimize the difference between the predicted future state and the inferred target state:$\mathcal{L}\_{\text{MPC}} = \|| s_{t+k} - s_g\||_2^2$.
>
> At each iteration, we sample a population of action sequences from a Gaussian distribution and update the mean and covariance using gradient descent, enabling the policy to reach the desired interaction outcome.  For PPO, we use the same reward formulation: $-\|| s_{t+k} - s_g\||_2^2$,
> and train a goal-conditioned policy to maximize expected return using standard PPO updates. [**For clarity**] We will add these to the second paragraph of Sec 3.2.
>
> ---
>
> > The use of Slot Attention is a critical component for learning object-centric representations. Given that the number of objects can vary across different environments, it would be helpful to know how the number of slots is determined. Conducting ablation studies on the effect of the number of slots on the model's performance in different environments would be better.
>
> **R5**: Thank you for the insightful comment. In our experiments, we heuristically set the number of slots to be slightly larger (typically +2 or +3) than the true number of objects present in the environment.
>
> This allows Slot Attention to stably learn object-specific representations, given its inherent limitations in dynamically adapting to varying object counts. And we think this heuristic is also practical in real-world settings where a rough estimate of object count is often available, either through prior domain knowledge or by incorporating auxiliary tools such as segmentation-based object proposal methods (e.g., SAM).
>
> While slot count selection is orthogonal to the core contributions of our method, we agree that it is helpful to include an ablation. Below, we report performance on the Gym-Fetch (Compositional generalization, 5 objects) task by varying the number of slots:
> | # Slots (relative to true object count) | Success Rate |
> |----------------------------------------|---------------|
> | Exact (0)                               | 0.72 ± 0.09   |
> | +1                                      | 0.75 ± 0.13   |
> | +2                                      | 0.86 ± 0.12   |
> | +3                                      | 0.88 ± 0.11   |
> | +4                                      | 0.81 ± 0.13   |
> | +5                                      | 0.75 ± 0.15   |
> | +10                                      | 0.69 ± 0.20   |
>
> From the results, we see that using **+2 to +4** slots yields stable and comparable performance. Setting the number of slots to exactly the number of objects can limit flexibility, making it harder to handle occlusions or background clutter. Too many slots may lead to redundancy or object fragmentation. Slight over-provisioning offers a good balance between expressiveness and stability.
>
> ---
>
> > The paper leverages pre-trained vision models like DINO-v2 and R3M to extract high-quality features. A clarification would be helpful as to whether these pre-trained models are fine-tuned during the world model learning stage.
>
> **R6**: We do not fine-tune the pre-trained vision models (DINO-v2 and R3M) during world model learning, which makes the framework efficient and flexible. Our goal is to build task-specific world models efficiently by leveraging the rich representations already captured by these models. [**For clarity**] We will clarify this point in the first paragraph of Section 3 in the camera-ready version.
>
> > The reconstruction loss is applied in the object-centric embedding space rather than the raw image space. The reasoning behind this choice could be further explained. While this approach is likely computationally efficient, it appears to heavily rely on the quality and generalization of the pre-trained model's features, which might lead to suboptimal learned representations if the pre-trained model's features are not sufficiently expressive for a given complex task.
>
> **R7**: Yes, we use reconstruction loss in the object-centric embedding space to reduce computational cost while focusing learning on semantically meaningful features. R3M, being pre-trained on robotic data, provides features well-suited for control tasks. DINO-v2 captures strong semantic representations of visual scenes (also validated in DINO-WM (Zhou et al., 2025)). We believe these pre-trained embeddings offer a sufficiently rich foundation, which our disentanglement module further refines to support downstream dynamics learning and decision-making.
>
>
> ---
> We hope these responses address your concerns. If so, we would be truly grateful if you would consider adjusting your rating accordingly. Thank you again for your time and thoughtful review.

---

> > ### Author Response · Authors · 2025-08-05
> > **Summary on the Rebuttal and Kind Reminder for Follow-ups**
> >
> > Dear Reviewer gBUX,
> >
> > Thank you again for your thoughtful review. In the rebuttal, we provided further clarifications and explanations on several technical points, including: the two-stage design (R3), the inverse mapping (R4), whether pre-trained models were fine-tuned (R6), and the rationale for using reconstruction in the feature space (R7). We also provided additional validations on the choice of the number of slots (R5), supported with new experiments.
> >
> > We also outlined executable camera-ready revision plans by specifying the exact sentences we would add or revise to improve clarity in the final version (see tags: [**For clarity**]). Regarding visualizations (R2), since file uploads are not allowed in the rebuttal, we will include visual results in the camera-ready version. In the meantime, we hope the probing analysis and LPIPS scores we provided can offer useful insights into the quality of the reconstructions.
> >
> > We hope these clarifications address your concerns. If you have any remaining questions or suggestions, we would be more than happy to follow up. Thank you once again for your valuable feedback!
> >
> > Best regards,
> >
> > Authors

---

> > > ### Comment · Reviewer_gBUX · 2025-08-06
> > >
> > > Thanks for the detailed responses. The authors' reply has largely resolved my concerns, and I am willing to increase my score by one point.

---

> > > > ### Author Response · Authors · 2025-08-06
> > > > **Thank you!**
> > > >
> > > > Thank you for your feedback! We are very glad that our response addressed your concerns, and we truly appreciate your willingness to raise the rating. We will make sure to include all the promised clarifications in the camera-ready version.

---

### Official Review · Reviewer_Kuio · 2025-07-02

**Clarity:** 3
**Significance:** 3
**Originality:** 3
**Rating:** 5
**Confidence:** 4

**Summary:**

This paper introduces FIOC-WM, a framework for object-centric model-based reinforcement learning that learns an object-centric world model and hierarchical policies in two separate stages: offline world model learning and online policy learning. By decomposing environment dynamics into object-level interactions and attribute-level states, world model is able to capture both static and dynamic properties. It begins by extracting object-centric representations directly from pixel observations using pre-trained visual encoders (DINO-v2, R3M) and the slot attention method (based on Videosaur), followed by variational autoencoders with contrastive losses to disentangle static and dynamic attributes. Leveraging this two-level factorization, a hierarchical policy framework is used: the high-level policy selects interaction graphs indicating which objects should interact, and the low-level policy executes these interactions using either MPC or PPO. The approach is evaluated on several robotic manipulation and embodied AI benchmarks (OpenAI Gym Fetch, iGibson, Libero), comparing against model-based RL methods (DreamerV3, TD-MPC2), model-based planning (DINO-WM) and prior object-centric RL baselines (FOCUS, EIT), FIOC-WM consistently achieves superior world model accuracy and better policy performance on most benchmarks.

**Questions:**

1. The paper mentions using both PPO and MPC frameworks for low-level policies. Are there any recommendations or insights on when to prefer PPO over MPC, or vice versa? If time permits, it would be valuable to see ablation exploring this comparison.
2. The paper claims that the method can handle long-horizon tasks. How long exactly? Would it be feasible for environments requiring thousands or even hundreds of thousands of steps? Additionally, does the performance degrade as the horizon length increases?
3. As stated in the original Videosaur paper, the quality of slot extraction tends to degrade with longer video lengths. Has this limitation affected the performance of FIOC-WM in your experiments?

**Ethical Concerns:**

["NO or VERY MINOR ethics concerns only"]

**Final Justification:**

The paper initially had issues with clarity and lacked details about the proposed method. However, these concerns were addressed in the authors’ rebuttal through additional experiments, and the authors have promised to resolve the remaining clarity issues in the camera-ready version. Therefore, I have increased the clarity score by one point.

**Limitations:**

Yes. Limitations are briefly mentioned in the section 6

**Paper Formatting Concerns:**

The paper shows no formatting issues.

**Quality:**

3

**Strengths And Weaknesses:**

##### Strengths:

1. Policy Performance Across Benchmarks. FIOC-WM outperforms or matches strong baselines (DreamerV3, TD-MPC2, DINO-WM, FOCUS, EIT) on several robotic benchmarks, especially in policy learning, demonstrating its practical advantage.
2. Generalization and Compositionality. The model shows robustness across attribute generalization, compositional tasks, and skill recomposition, highlighting its strength in tasks involving unseen combinations of known components.
3. The paper is well written, with some clarity remarks, listed in Weaknesses section.


##### Weaknesses:

1. There is no visualization of the slot mappings to the environment observations. Including such visualizations would help clarify that the method's performance is driven by object-centric representations rather than unstructured data. For example, all important objects might otherwise be mapped to a single slot.
2. Some essential details are missing from the paper. The paper notes that either MPC or PPO could be used for low-level policies in the second step, but does not indicate which was used in the experiments. Clarifying these points would improve transparency and reproducibility.
3. The architecture is difficult to understand. Including schematic diagrams of the dynamics model and the attributes VAE would greatly improve clarity and help to better understand the structure of the world model.

4. In Appendix E.3, it appears that the Videosaur training time is not included. For the sake of fairness, it would be important to account for the slots attention training time as well, particularly since it needs to be trained separately for each environment.

5. It seems that Table A4 is missing the hidden dimensions for the MLP decoder used in Videosaur.

---

> ### Author Rebuttal · Authors · 2025-07-31
>
> We thank you for the detailed and constructive review, and we appreciate your recognition of our model contributions and writing. Most of the questions pertain to technical details, which we address through clarifications and additional experiments provided below. In the camera-ready version, we will incorporate these clarifications and include visualizations at the pixel level. We hope these responses address your concerns.
>
> ---
>
> > There is no visualization of the slot mappings to the environment observations.
>
> **R1**: We agree that visualizing slot-to-object mappings would help clarify the effectiveness of our object-centric representations. Since in the rebuttal we cannot add figures anymore, we will not be able to provide them here, but in the camera-ready version, we will include such visualizations (e.g., reconstructed images and slot masks).
>
> While our current evaluation focuses on probing results (Fig. 5(a)) to show disentanglement of factors like position, velocity, and color, these fine-grained results suggest that object-level separation is also achieved. This is further supported by the use of the Videosaur backbone, which has shown strong object-wise disentanglement in prior work. That said, we acknowledge that a direct qualitative assessment would strengthen this point, and we will ensure that these visualizations are included in the final version.
>
> [1] Zadaianchuk, Andrii, Maximilian Seitzer, and Georg Martius. "Object-centric learning for real-world videos by predicting temporal feature similarities." NeurIPS 2023.
>
> ---
>
> > Some essential details are missing from the paper. The paper notes that either MPC or PPO could be used for low-level policies in the second step, but does not indicate which was used in the experiments.
>
> **R2**: Thank you for pointing this out. We apply MPC for the Gym-Fetch and Franka-Kitchen environments, and PPO for LIBERO and iGibson tasks. We will add this clarification after Line 220 in the revision.
>
> Additionally, we conducted an ablation study to compare the use of PPO and MPC across all environments. The results, which show comparable final success rates, confirm that both approaches are feasible. We will include this ablation table in Appendix D.
>
> |Environment|MPC|PPO|
> |-|-|-|
> |Gym Fetch (Task 1)|0.95±0.03|0.96±0.03|
> |Gym Fetch (Task 2)|0.98±0.01|0.92±0.05|
> |Franka Kitchen (Task 1)|0.82±0.04|0.79±0.06|
> |Franka Kitchen (Task 2)|0.79 ± 0.06|0.76±0.03|
> |i-Gibson (Task 1)|0.72±0.09|0.76±0.12|
> |i-Gibson (Task 2)|0.74±0.13|0.72±0.10|
> |Libero (Task 1)|0.73±0.05|0.81±0.11|
> |Libero (Task 2)|0.76±0.06|0.74±0.09|
>
> ---
>
> > The architecture is difficult to understand. Including schematic diagrams of the dynamics model and the attributes VAE would greatly improve clarity and help to better understand the structure of the world model.
>
> **R3**: Thank you for the suggestion. The dynamics model and attribute VAE are currently shown as modules in Fig. 3. To improve clarity, we will include two additional schematic diagrams illustrating their input/output flow and network structures in Appendix D of the camera-ready.
>
> For reference, the detailed loss functions are provided in Section 3.1, and the corresponding hyperparameters are listed in Appendix E.1.
>
> ---
>
> > In Appendix E.3, it appears that the Videosaur training time is not included. For the sake of fairness, it would be important to account for the slots attention training time as well, particularly since it needs to be trained separately for each environment.
>
> **R4**: Thank you for the suggestion. You are right that for a fair comparison, the training time of the Slot Attention (Videosaur) module should be reported. We have now included the training times for each environment (all measured on an A100 GPU with 40GB memory):
> - Sprites-World: 1 hour
> - Fetch: 4 hours
> - iGibson: 5 hours
> - Libero: 5 hours
> - Franka-Kitchen: 4.5 hours
>
> We will add this information to Appendix E.3 in the camera-ready version.
>
> ---
>
> > It seems that Table A4 is missing the hidden dimensions for the MLP decoder used in Videosaur.
>
> **R5**: Thank you for catching this. The hidden dimensions for the MLP decoder in Videosaur were omitted from Table A4. We use:
> - iGibson, Libero, Franka-Kitchen: 4-layer MLP with 1024 hidden units
>
>
> - Fetch: 3-layer MLP with 512 hidden units
>
>
> - Sprites-World: 2-layer MLP with 256 hidden units
>
>
> These settings are chosen based on the complexity of each environment. We will update Table A4 accordingly in the camera-ready version.
>
> ---
>
> > The paper mentions using both PPO and MPC frameworks for low-level policies. Are there any recommendations or insights on when to prefer PPO over MPC, or vice versa? If time permits, it would be valuable to see ablation exploring this comparison.
>
> **R6**: We choose MPC for tasks with relatively simple semantics and shorter horizons (e.g., Gym-Fetch, Franka-Kitchen), and PPO for more complex environments involving long-horizon goals and rich semantics (e.g., LIBERO, iGibson). This decision balances efficiency and learning flexibility. As mentioned previously, we also include an ablation comparing PPO and MPC across environments, which shows that both are feasible with similar final performance. We will add these insights and the corresponding table in Appendix D.
>
> ---
>
> > The paper claims that the method can handle long-horizon tasks. How long exactly? Would it be feasible for environments requiring thousands or even hundreds of thousands of steps? Additionally, does the performance degrade as the horizon length increases?
>
> **R7**: By “long-horizon,” we refer to the method’s ability to solve tasks with complex temporal structure by decomposing them into meaningful sub-tasks using learned interaction structures and disentangled features. The actual episode length depends on when the agent completes the task, rather than being fixed.
> While we have not explicitly tested environments requiring hundreds of thousands of steps, our hierarchical framework is designed to handle increasing task complexity through temporal abstraction. To support this, we evaluate variants of the Franka-Kitchen environment with different numbers of sub-tasks. For example:
> - Task 1: Open the microwave → Move the kettle → Turn on the stove → Turn on the light (4 sub-tasks)
> - Task 2:  Open the microwave → Turn on the light → Slide the cabinet → Open the cabinet → Turn on the top burner (5-subtasks)
> - Task 2:  Open the microwave →  Move the kettle → Turn on the light → Turn on the stove → Open the cabinet → Turn on the top burner  (6-subtasks)
>
>
> | # Sub-tasks        | Ours (FIOC)      | TD-MPC2        |
> |-------------|-----------------|----------------|
> | 4           | **0.82 ± 0.04**  | 0.83 ± 0.03    |
> | 5           | **0.76 ± 0.02**  | 0.72 ± 0.05    |
> |  6         | **0.70 ± 0.02**  | 0.61 ± 0.03    |
>
> These results show that while both methods perform comparably on moderate tasks (4 sub-tasks), **FIOC outperforms TD-MPC2 (best-performing baseline for this task) as task complexity increases**, showing better scalability and robustness for long-horizon tasks. This supports our claim that the hierarchical design and object-centric representations in FIOC enable more effective temporal abstraction.
>
> ---
>
> > As stated in the original Videosaur paper, the quality of slot extraction tends to degrade with longer video lengths. Has this limitation affected the performance of FIOC-WM in your experiments?
>
> **R8**: In our experiments, we did not observe significant degradation in slot extraction quality with longer video lengths. This is likely because the environments we consider are not as semantically complex or visually diverse as the long-form natural videos (e.g., YouTube-VIS) used in the original Videosaur paper. Most robotic control and embodied AI environments involve more structured and repetitive dynamics. That said, we acknowledge this known limitation of Videosaur and will add a note in Appendix E.1 stating: “*While slot extraction quality may degrade over very long or highly diverse videos, we did not observe such issues in the structured environments used in our experiments.*”
>
> ---
>
> Thank you again for your positive feedback and insightful questions! Please let us know if you have any further questions!

---

> > ### Comment · Reviewer_Kuio · 2025-08-05
> >
> > Thank you for addressing my questions and providing additional experiments. I will keep my score as is. While this may not be a highly impactful paper for the broader machine learning and AI research community, I believe it is an important contribution to the object-centric learning research field.

---

> ### Author Response · Authors · 2025-08-05
> **Thank you!**
>
> Thank you so much for your response and your positive feedback! We are glad that we were able to address your concerns, and we truly appreciate your recognition of the paper’s contribution to the object-centric learning community.
>
> We believe this work can also provide useful insights for broader areas such as (object-centric) representation learning, world models, and RL/control. Also, as mentioned, we will make sure to incorporate all the promised revisions in the camera-ready version.

---

### Official Review · Reviewer_VDhr · 2025-07-02

**Clarity:** 2
**Significance:** 2
**Originality:** 3
**Rating:** 5
**Confidence:** 3

**Summary:**

This paper proposes the FIOC-WM  framework for learning structured world models that capture both object-level representations and their interactions from high-dimensional visual observations. The core idea is to factor the world model state space across individual objects, decomposing each object into dynamic (time-varying) and static (time-invariant) attributes via DINO-v2, R3M and Slot Attention, and modeling their interactions as sparse, time-varying graphs. FIOC-WM consistently outperforms baselines like DreamerV3, TD-MPC2, and DINO-WM across metrics such as LPIPS, success rate, and generalization gap.

**Questions:**

1.  Is the same offline training setup used for baseline models such as DreamerV3 and TD-MPC2? Please clarify whether DreamerV3 and TD-MPC2 are also trained in the offline setting using the same dataset as FIOC-WM

2. You mention using both Model Predictive Control (MPC) and Proximal Policy Optimization (PPO) for training the low-level policy. Under what conditions is each method used?
   * When PPO is applied, is it trained using imagined rollouts from the learned world model (i.e., model-based PPO)?
   * How is the action sampling process implemented when using MPC with the Cross Entropy Method (CEM)? Specifically:

     * What are the hyperparameters (e.g., number of gradient descent steps, population size)?
     * Could you provide implementation details or pseudocode of the policy training?

3.  How is the prior distribution $P(G)$ over graph structures defined?  What learning procedure is used to infer the graph structures? This need to be clarified.

4. Equation (6) defines the reward prediction loss, but it is unclear how this learned reward model is utilized during policy training or planning. Is the predicted reward used in MPC rollout evaluation or in PPO optimization?

**Ethical Concerns:**

["NO or VERY MINOR ethics concerns only"]

**Final Justification:**

Weaknesses 4, as well as Questions 1-4 and the limitation, have been addressed in the rebuttal. However, Weakness 1, 2, 3 remains unaddressed.

**Limitations:**

The model assumes a fixed number of object slots. It is unclear how it generalizes to environments where the number of objects changes, or where previously unseen objects appear.

**Quality:**

3

**Strengths And Weaknesses:**

Strengths:

1. The paper proposes a principled factorization of the state representation into dynamic (time-varying) and static (time-invariant) components, and learns their respective dynamics accordingly. This decomposition is intuitive and aligns well with the physical properties of real-world environments.

2. The use of sparse, time-varying graphs to model interactions between objects provides a structured and interpretable inductive bias for object-centric model-based reinforcement learning, enabling more accurate prediction of environment dynamics.

3. The hierarchical policy design—where the high-level policy plans over interaction graphs and the low-level policy executes them—effectively leverages the compositional nature of learned object interactions, resulting in improved performance on long-horizon and generalization tasks.

Weaknesses:

1. While the predictive quality of future observations is quantitatively evaluated using the LPIPS metric, the paper does not provide qualitative visualizations comparing long-term predictions between FIOC-WM and baseline methods. Including side-by-side image rollouts (e.g., from Table A3) would help illustrate in which scenarios FIOC-WM makes accurate predictions and where baselines fail. Such visual comparisons are critical to support the claim of improved long-term predictive fidelity.

2. The paper introduces a surrogate variable ut to model the distribution over interaction graphs given the latent state st, but the role and interpretation of this variable remain unclear. The learning process for the interaction graph, including how ut  is optimized and how it influences the transition dynamics, is under-explained and would benefit from additional clarification.

3. The motivation for adopting two specific approaches—(i) variational mask learning and (ii) conditional independence testing—for learning the state transition distribution is not sufficiently justified. It is unclear why these methods were chosen over other alternatives or what specific advantages they offer in the context of object-centric interaction modeling.

4. Although FIOC-WM shows better performance than baselines in generalization tasks (e.g., Fig. A3), it benefits from pre-trained visual encoders (DINO-v2, R3M), while most baselines do not. This raises concerns about the fairness of the comparison. It remains unclear how much of the performance gain stems from the proposed factorized world model itself versus the use of strong pre-trained features. An ablation comparing against DreamerV3 or TD-MPC2 with access to the same pre-trained encoder would help isolate the contribution of FIOC-WM.

---

> ### Author Rebuttal · Authors · 2025-07-31
>
> Thank you for your careful review. We appreciate the detailed comments and your recognition of our model. Most of the concerns are centered around technical details, and we provide clarifications and additional evaluation results for each point below.
>  We hope these responses address your concerns. For points requiring clarification, we outline concrete plans for how they will be incorporated into the camera-ready version. The additional evaluations further support the soundness of our technical design. Please let us know if you have any remaining questions or concerns.
>
> ---
>
> > No qualitative visualizations comparing long-term predictions between FIOC-WM and baseline methods.
>
> **R1**: We agree that qualitative visualizations of pixel observations would be useful. Since in the rebuttal we cannot add figures anymore, we will not be able to provide them here, but in the camera-ready version, we will include side-by-side image rollouts in the appendix to compare long-term predictions made by FIOC-WM and baseline methods.
>
> ---
>
> > Role of [ $u_t$] and clarification of learning process
>
> **R2**: The $\mathbf{u}_t$ variable represents the latent parameters of the distribution from which the interaction graph is sampled at each timestep, which allows us to model the fact that the interaction graphs (and their distributions) are time-varying.
>
> Specifically, for each object pair $(i, j)$ at time $t$, we encode their latent states $s_t^i$ and $s_t^j$ using an RNN-based encoder to obtain a pairwise embedding:
> $$
> \mathbf{u}_t^{ij} = f\_{\text{enc}, \phi_u}(s_t^i, s_t^j)
> $$
> (as described in Equation (A2)).
>
> We then use these embeddings to infer the graph structure $G_t$ in one of two ways:
>
> 1. **Sampling edges** using a Gumbel-Softmax distribution over $\mathbf{u}_t$ (Equation A3), or
>
> 2. **Quantizing** $\mathbf{u}_t$ into a discrete codebook of interaction types (Equation A4), followed by decoding into a graph via:$G_t \sim g\_{\text{dec}}(e_z)$, where $e_z$ is the prototype vector.
>    (as shown in Equation (A5)).
> In the camera-ready version, we will put the above explanation into the main paper as we have one extra page available.
>
> ---
>
> > prior distribution P(G) over graph structures defined? What learning procedure is used to infer the graph structures?
>
> **R3**: The prior distribution $P(G)$ is defined as an independent Bernoulli distribution over each edge:
> $
> G_{ij} \sim \text{Bernoulli}(p),$ where $p \in (0, 1)$ controls the sparsity level. We set $p \ll 0.5$ (e.g., $p = 0.1$) to encourage sparse and interpretable interaction graphs by biasing toward no-edge configurations.
>
> In the camera-ready, we will clarify this in Section 3.1 (another paragraph after Line 151) by explicitly describing the sparsity prior and summarizing the graph inference procedure from state encoding to discrete graph generation.
>
> ---
>
> > The motivation for adopting two specific approaches—(i) variational mask learning and (ii) conditional independence testing—for learning the state transition distribution is not sufficiently justified.
>
> **R4**: We adopt variational masking and CIT because both are well-established structure learning strategies. Variational methods benefit from scalability and can be efficiently trained via likelihood matching, while CIT offers statistical grounding for inferring dependency structures. We also note that, due to the flexibility of our framework, it can accommodate any off-the-shelf structure learning method.
> To further support our design choice, we additionally evaluated Dynotears [1], a strong baseline for causal structure learning [2]. We use it on the Sprites-World environment with 3, 5, and 9 objects. The results are shown below:
> | # Objects | Variational (Cat.) | Variational (Code.) | CIT           | Dynotears     | Attention-based |
> |-----------|---------------------|----------------------|---------------|---------------|------------------|
> | 3         | 0.09        | 0.08      | 0.06 | 0.09   | 0.12    |
> | 5         | 0.12        | 0.15      | 0.13     | 0.15   | 0.20      |
> | 9         | 0.27         | 0.31      | 0.35     | 0.36   | 0.41      |
>
> The results show that DynoTears performs similarly to CIT, but **worse than variational methods** across all object counts. This supports our use of variational masking and CIT as effective and principled choices for object-centric interaction modeling. To clarify our choice, we will add more results for DynoTears for other settings in the Appendix of the revised paper.
>
> [1] Pamfil, Roxana, et al. "Dynotears: Structure learning from time-series data."AISTATS 2020.
>
> [2] Vowels, Matthew J., Necati Cihan Camgoz, and Richard Bowden. "D’ya like dags? a survey on structure learning and causal discovery." ACM Computing Surveys 55.4 (2022): 1-36.
>
> ---
>
> > Pre-trained models for Dreamer and TD-MPC
>
> **R5**: Thank you for the suggestion. We originally did not include this comparison because Dreamer-V3 is designed to reconstruct pixel-level observations, while TD-MPC2 predicts actions, terminal flags, and rewards directly from raw pixels. However, to better isolate the contribution of our proposed factorized world model from the benefits of pre-trained visual features, we implemented a fairer comparison.
>
> Specifically, we adapted both baselines to use DINO embeddings as input:
> - For Dreamer-V3, we replaced the pixel encoder with a frozen DINO encoder and used DINO-WM's decoder to reconstruct the observations.
> - For TD-MPC2, we used DINO features to predict actions, terminal values, and rewards via its original decoding heads.
>
>
> The updated results are shown below:
>
> |Envs | FIOC | Dreamer-V3 (DINO/original)|TD-MPC2 (DINO/original)|
> |-|-|-|-|
> |Kitchen|0.82|0.77/0.75|0.79/0.83|
> |i-Gibson|0.76|0.71/0.69|0.73/0.72|
> |Libero|0.81| 0.69/0/.65|0.74/0.76|
>
> Using pre-trained DINO features improves both Dreamer-V3 and TD-MPC2 performance in some cases, but they still underperform compared to FIOC. This suggests that while strong visual representations help, our proposed **factorized world model and hierarchical policy structure** are key contributors to the performance gain.
>
> ---
>
> >  Is the same offline training setup used for baseline models such as DreamerV3 and TD-MPC2?
>
> **R6**: Yes, all are trained on the same dataset as we mentioned in Table A6.
>
> ---
>
> >  You mention using both MPC and PPO for training the low-level policy. Under what conditions is each method used?
>
> **R7**: We use **variational masks** for the results reported in Table 2. Specifically, for all methods (excluding those evaluated under the nSHD metric), we adopt variational masks to infer the interaction structures. For downstream control, we apply **MPC** for the Gym-Fetch and Franka-Kitchen environments, and use **PPO** for LIBERO and iGibson tasks.
> We also conducted an ablation study to evaluate the impact of using MPC or PPO on all environments. The results of this ablation are shown below. The results indicate that using either PPO or MPC is feasible with similar final success rates. We will add this Table in Appendix D.
> |Environment|MPC|PPO|
> |-|-|-|
> |Gym Fetch (Task 1)|0.95±0.03|0.96±0.03|
> |Gym Fetch (Task 2)|0.98±0.01|0.92±0.05|
> |Franka Kitchen (Task 1)|0.82±0.04|0.79±0.06|
> |Franka Kitchen (Task 2)|0.79 ± 0.06|0.76±0.03|
> |i-Gibson (Task 1)|0.72±0.09|0.76±0.12|
> |i-Gibson (Task 2)|0.74±0.13|0.72±0.10|
> |Libero (Task 1)|0.73±0.05|0.81±0.11|
> |Libero (Task 2)|0.76±0.06|0.74±0.09|
>
> ---
>
> > When PPO is applied, is it trained using imagined rollouts from the learned world model (i.e., model-based PPO)?
>
> **R8**:  Yes, we used both the imagined rollouts and the rollouts from the replay buffer to learn the PPO policy.
>
> ---
>
> > How is the action sampling process implemented when using MPC with the Cross Entropy Method (CEM)? What are the hyperparameters (e.g., number of gradient descent steps, population size)?
>
> **R9**: We implement MPC using the Cross Entropy Method (CEM) with the following hyperparameters:
> - CEM iterations: 5 for Fetch, Franka, and i-Gibson; 8 for Libero
> - Population size: 500 for Fetch, 1000 for others
> - Planning horizon: 10 for Fetch, 20 for Franka, i-Gibson, and Libero
> We will add these in the camera-ready Appendix E.2 as an individual table and include an algorithmic pseudocode as well in the appendix.
>
> ---
>
> > Eq6 defines the reward prediction loss, but it is unclear how this learned reward model is utilized during policy training or planning. Is the predicted reward used in MPC rollout evaluation or in PPO optimization?
>
> **R10**: The reward prediction model is used during Stage 1 for learning the world model. Specifically, we supervise it using ground-truth rewards from the offline dataset, following the same setup as in Dreamer-v3 and TD-MPC2.
>
> >  The model assumes a fixed number of object slots. It is unclear how it generalizes to environments where the number of objects changes, or where previously unseen objects appear.
>
>  **R11**: To assess the model's ability to generalize to a greater number of objects, we train on environments with 3 objects and evaluate on tasks involving 6 and 8 objects with the fixed world model. As shown in the table below, FIOC achieves strong generalization performance, comparable to or better than baselines like EIT, and outperforms Dreamer-V3 and TD-MPC2 under distribution shifts in object count.
>
> | Envs        | FIOC (Ours) | Dreamer-V3 | EIT   | TD-MPC2 |
> |-------------|-------------|------------|-------|---------|
> | 3 objects   | 0.93        | 0.96     | 0.94  | **0.97** |
> | 6 objects   | **0.81**      | 0.54       | 0.77 | 0.62    |
> | 8 objects   | **0.70**    | 0.44       | 0.62  | 0.53    |
>
> ---
>
> We hope these can address the concerns. Please let us know if you have any further questions. Thank you again for your time and efforts!

---

> > ### Comment · Reviewer_VDhr · 2025-08-05
> >
> > Thank you for your response. I still have the following concerns:
> >
> > What is the mixing ratio between model-generated rollouts and replay buffer data in your PPO training? Would PPO still converge if it were trained solely on data from the world model rollouts?
> >
> > How does your method generalize from training on 3 objects to handling 6–8 objects during deployment? As far as I understand, the model is only exposed to graphs with 3 objects during training. What is the complete inference process at deployment time? Which components are involved, and how exactly is generalization achieved? In particular, how does the fact that the graph structure at inference time differs from that seen during training affect performance?
> >
> > Why does combining TDMPC with DINO lead to worse performance? Could you provide insights into the cause of this degradation?

---

> > > ### Author Response · Authors · 2025-08-05
> > > **Further Response (1/2)**
> > >
> > > Thank you for your feedback and further questions. We provide the response below:
> > >
> > > ---
> > >
> > > > We empirically set the mixing ratio between model-generated rollouts and replay buffer data to **1:1**, though we did not perform extensive tuning on this hyperparameter.
> > >
> > > To assess the effect of relying solely on model-generated data, we did conduct an ablation study on the i-Gibson environment before:
> > >
> > > | Environment         | 1:1 Ratio         | Full Generated Only |
> > > |---------------------|------------------|----------------------|
> > > | i-Gibson (Task 1)   | 0.76 ± 0.12       | 0.65 ± 0.21          |
> > > | i-Gibson (Task 2)   | 0.72 ± 0.10       | 0.58 ± 0.17          |
> > >
> > > Despite the increased complexity of i-Gibson, PPO still learns meaningful behavior using only generated rollouts, although performance is lower than with a mixed ratio. This suggests that while full reliance on model-generated data is possible, incorporating real data improves performance and stability.
> > >
> > > We plan to include this ablation in the camera-ready version and are running similar analyses on the remaining environments. Note that for all baselines, we ensure a consistent amount of generated and replay data to maintain fair comparisons.
> > >
> > > > How does your method generalize from training on 3 objects to handling 6–8 objects during deployment? As far as I understand, the model is only exposed to graphs with 3 objects during training. Which components are involved, and how exactly is generalization achieved?
> > >
> > > Great question! First, most components of the world model are shared between training and deployment, regardless of the number of objects. Specifically, the following components are directly transferable:
> > >
> > > - Observation encoder: $\hat{o}\_t \leftarrow q\_\phi(o_t)$
> > > - Latent state inference: $s_t^i \leftarrow q_\phi(s_t^i \mid \hat{o}_t)$
> > > - Latent state factorization: $s_t^i = (d_t^i, c_t^i)$
> > > - Reward model: $p_r(r_t \mid s_t, a_t)$
> > > - State Transition model: $p_s(s_{t+1} \mid s_t, a_t, G_t)$
> > >
> > > These modules generalize across different numbers of objects and can be used directly at deployment.
> > >
> > > For the interaction graph $P_u (G_t \mid s_t)$, we fine-tune it using a small number of rollouts (approximately 10% of the original training data; the pair-wise state network can be fixed). The low-level policy is also transferable since it focuses on pairwise object interactions, which are structurally similar even with more objects.
> > >
> > > Thus, generalization is primarily supported by the **world model and the low-level policy**, while only the interaction graph and the high-level policy (scheduler) require adaptation.
> > >
> > > > What is the complete inference process at deployment time?
> > >
> > > As listed above, the inference process at deployment involves both reusing trained components and adapting others as needed:
> > >
> > > 1. **Load pretrained parameters** for:
> > >    - Observation encoder: $q_\phi(o_t)$
> > >    - Latent state inference: $q_\phi(s_t^i \mid \hat{o}_t)$
> > >    - Transition model: $p_s(s_{t+1} \mid s_t, a_t, G_t)$
> > >    - Reward model: $p_r(r_t \mid s_t, a_t)$
> > >    - Low-level policy: $\pi^l$
> > >
> > > 2. **Collect a small number of rollouts (∼10%)** in the new environment with more objects.
> > >
> > > 3. **Fine-tune the interaction graph model** $p_u(G_t \mid s_t)$ based on the collected rollouts.
> > >
> > > 4. **Fine-tune the high-level policy $\pi^h$ online**.
> > >
> > > > In particular, how does the fact that the graph structure at inference time differs from that seen during training affect performance?
> > >
> > > Since both the **world model** and the **low-level policy** operate on local or pairwise object interactions, they are largely robust to changes in the global graph structure. As a result, they are not significantly affected by the increased complexity of the interaction graph at test time.
> > >
> > > In our setup, the primary difference in graph structure arises from the increased number of objects: e.g., during deployment, the model may encounter scenes with 6–8 objects compared to 3 during training. Naturally, this results in denser and more complex (different) graphs at inference time. Despite this structural shift, our results show that the system continues to perform well, indicating that the **world model** and **low-level policy** generalize effectively, while the **interaction graph** and **high-level policy** can be adapted with minimal fine-tuning.

---

> > > > ### Author Response · Authors · 2025-08-05
> > > > **Further Response (2/2)**
> > > >
> > > > > Why does combining TDMPC with DINO lead to worse performance? Could you provide insights into the cause of this degradation?
> > > >
> > > > We believe the performance degradation arises from a mismatch between the design of the TD-MPC framework and the characteristics of DINO representations. TD-MPC and its variants are designed to predict actions, terminal values, and rewards or return, rather than reconstructing raw pixels or learning semantically rich visual features.
> > > >
> > > > While DINO provides strong pretrained embeddings that capture high-level visual semantics, these representations may contain redundant or entangled information that is less structured or less aligned with the predictive objectives of TD-MPC. This redundancy can make optimization difficult and make it harder for the model to extract actionable, task-relevant signals.
> > > >
> > > > In fact, this observation supports one of our core motivations: strong pretrained representations often require **further disentanglement** to become effective for reinforcement learning and planning tasks. Our framework is explicitly designed to structure and refine these embeddings for downstream control.
> > > >
> > > > ---
> > > >
> > > > We hope these responses can address your remaining questions and concerns. Please do not hesitate to let us know if anything remains unclear. We are grateful for your insightful points, which really contribute meaningfully to improving the work.

---

> > > > > ### Comment · Reviewer_VDhr · 2025-08-05
> > > > >
> > > > > Why does fine-tuning the graph model lead to improved performance? What assumptions are made about the object interactions in the training and test sets? Is it possible that the test set contains object interactions that were never seen during training? If so, what new capabilities does the graph model acquire through fine-tuning, and what prior knowledge from the training data enables it to adapt with fewer samples?
> > > > >
> > > > > Furthermore, why does increasing the number of objects not affect the behavior of your low-level policy? What is the nature of the task being solved? What roles do the low-level and high-level policies play, respectively? As the number of objects increases, the environment dynamics should become more complex—why does this change in dynamics not impact the behavior of the low-level policy? Please illustrate this with specific examples from your experiments.
> > > > >
> > > > > Additionally, are all the objects in your environment identical, differing only in quantity? What happens if the new objects are entirely different from those seen during training?

---

> > > > > > ### Author Response · Authors · 2025-08-05
> > > > > > **Further Response on Generalization Tasks**
> > > > > >
> > > > > > Thanks for the further questions!
> > > > > >
> > > > > > > Why does fine-tuning the graph model lead to improved performance?
> > > > > >
> > > > > > The primary reason is that, in the new setting, the **number of objects changes**, which alters the input size to the graph inference module. While the underlying **pairwise interaction network** remains unchanged, the model must adapt to reasoning over a different number of objects and possibly more complex graph structures.
> > > > > >
> > > > > > Fine-tuning with a small number of rollouts allows the model to adjust to these structural differences. The setup is also realistic in practical applications, where a few demonstrations or rollouts in the new environment are typically available.
> > > > > >
> > > > > > > What assumptions are made about the object interactions in the training and test sets? Is it possible that the test set contains object interactions that were never seen during training? If so, what new capabilities does the graph model acquire through fine-tuning, and what prior knowledge from the training data enables it to adapt with fewer samples?
> > > > > >
> > > > > > We assume that the **type** of objects (e.g., bottles versus balls) is similar between the training and test environments. During testing, we vary **attributes** such as color and mass (this supports the generalization on previously unseen object combinations).
> > > > > >
> > > > > > It is indeed possible that the test set contains **entirely new object categories**. In such cases, the model may encounter new object dynamics and interaction patterns. However, the interaction network is trained to capture generalizable interaction structures, so it can often adapt to these new combinations with minimal fine-tuning, especially when the dynamics are similar to those encountered during training.
> > > > > >
> > > > > > For example, in the Fetch setting, while the training environment may include only cube-shaped objects, the test environment could include switches or other shapes. As long as the new objects exhibit similar underlying dynamics, the **world model does not require retraining**, and the graph module can adapt with a small number of rollouts.
> > > > > >
> > > > > > In more extreme cases, e.g., where the test set introduces **entirely new dynamics** not present in the training data, retraining the dynamics model for that type of new object may be necessary. However, in most realistic settings where dynamics are shared and only superficial properties differ, the model can generalize effectively, leveraging its prior knowledge and fine-tuning the graph module to capture new interaction patterns.
> > > > > >
> > > > > > > Furthermore, why does increasing the number of objects not affect the behavior of your low-level policy? What is the nature of the task being solved? What roles do the low-level and high-level policies play, respectively? As the number of objects increases, the environment dynamics should become more complex—why does this change in dynamics not impact the behavior of the low-level policy? Please illustrate this with specific examples from your experiments.
> > > > > >
> > > > > > Great questions!! The key reason the low-level policy remains largely unaffected by the number of objects is that it is designed to operate on **localized, pairwise interactions**. In our framework, each **sub-goal** typically involves manipulating a specific object or coordinating between a **pair of objects**, which makes the control policy inherently **modular and reusable** across different object counts.
> > > > > >
> > > > > > As the number of objects increases, the overall environment dynamics indeed become more complex. However, this added complexity is primarily handled by the **high-level policy**, which plans the sequence and schedule of sub-goals (i.e., which object(s) to manipulate and in what order) based on the inferred interaction graph. The **low-level policy**, by contrast, focuses on executing individual manipulation sub-tasks and thus remains focused on local interactions.
> > > > > >
> > > > > > For example, in the **n-Push** environment, the agent needs to sequentially push multiple blocks to designated target locations. Regardless of whether there are 3 or 6 blocks, the low-level policy’s role in pushing a single block (or coordinating between a pair of blocks) stays the same. The additional complexity from more objects, e.g., needing to avoid obstacles created by other blocks, could be accounted for by the high-level planner, which schedules the sub-goals to avoid such conflicts.
> > > > > >
> > > > > > > Additionally, are all the objects in your environment identical, differing only in quantity? What happens if the new objects are entirely different from those seen during training?
> > > > > >
> > > > > > We consider the objects with different attributes but with the same type, like cubes and switches differ in color (fetch), size, and texture (in Libero).
> > > > > >
> > > > > > ---
> > > > > >
> > > > > > We hope these responses address your further questions! These were all excellent points, and we are truly grateful for the thoughtful discussion!

---

> > > > > > > ### Comment · Reviewer_VDhr · 2025-08-05
> > > > > > >
> > > > > > > Thanks for your response. Most of my concerns have been addressed, and I am willing to raise my score.

---

> > > > > > > > ### Author Response · Authors · 2025-08-05
> > > > > > > > **Thank you!**
> > > > > > > >
> > > > > > > > We are truly glad that we could address your questions! They are all very insightful and meaningful questions. We also sincerely appreciate your willingness to consider raising the score. We will make sure to incorporate all the revisions we mentioned into the camera-ready version. Thanks again for your thoughtful feedback and support!

---

### Official Review · Reviewer_E9iB · 2025-07-02

**Clarity:** 1
**Significance:** 2
**Originality:** 2
**Rating:** 4
**Confidence:** 3

**Summary:**

This work proposes a two-stage method called FIOC. First, authos propose to learn object-centric world model to model relationships between objects explicitly. Then, a low-level policy is trained to achieve desired interactions, and a high-level policy is trained to optimize rewards and supply interaction subgoals to the lower level policy. The authors then test FIOC on a variety of tasks, including SpriteWorld and robotic manipulation tasks, and demonstrate impressive performance.

**Questions:**

- Q1: How do you get the ground truth interaction structures G to get results shown Figure 6? Do I understand correctly that you only do that for SpriteWorld, and that ground truth G is not required unless you want to measure SHD? Also, how exactly is G represented? Is it an NxN matrix of values between 0 and 1? Or is it discrete?
- Q2: In the online stage, do you also update the world model with the new data you collect?
- Q3: Do you train the low-level policy fully inside the learned world model or using online observations? (Q4 and Q3 could be answered if you show the algorithms of different stages of your method)
 - Q4: In line 173, you say you only focus on subsets of objects with size of less than 2. Is that correct? Do you only focus on sets of single objects?
- Q5: FIOC considers two options for low-level policy: MPC and RL, and two options for graph inference: Variational and CIT. What version do you use in Table 2 results?

**Ethical Concerns:**

["NO or VERY MINOR ethics concerns only"]

**Final Justification:**

The paper was unclear to me when I first read it, but after the discussion with the authors I appreciate the idea more. I think the presented method is interesting, I find the hierarchical decomposition of interactions to be a promising direction, and therefore believe the paper should be accepted. That said, I still have some concerns regarding the presentation clarity, although the authors promised to address these in the final version.

**Limitations:**

yes

**Paper Formatting Concerns:**

no concerns

**Quality:**

2

**Strengths And Weaknesses:**

###### Strengths
- This work proposes an effective way to use observation factorization and object relationship graphs to achieve good performance.
- The performance is impressive. In particular, the boost in performance on generalization showcases the benefit of explicitly modelling the objects and their relationships;

###### Weaknesses
- The writing is not clear at all: I struggled to understand what G means exactly and how it's learned, and many other things are unclear as well. See questions.

Due to writing, I do not think this paper shoul dbe accepted. However, I find results quite compelling, and I'm willing to increase my score if authors address my concerns.

###### Typos and nitpicks
Figure 2 caption: gray node -> gray nodes
112: ave -> have, the sentence doesn't make sense grammatically
139: a observation -> an observation
292: fina -> final
Figure 6: shaded and non-shaded areas look confusing, I'd choose another color/pattern.

---

> ### Author Rebuttal · Authors · 2025-07-31
>
> Thank you for your thoughtful review and feedback. We sincerely appreciate your suggestions and your recognition of our methodology and performance.
>
> Most of the concerns relate to writing clarity and a few technical details. For clarity, we propose concrete and executable revisions that can be addressed through targeted clarifications in the camera-ready version. For the technical concerns, we provide essential clarifications and additional evaluations. We hope our responses adequately address your questions, and please let us know if you have any further concerns.
>
> ---
>
> > what G means exactly and how it's learned
>
> **R1**: We define $\mathcal{G} =$ { $G_1, \ldots, G_T$ } as the sequence of **interaction graphs** that represent the relational structure among objects over time (*Lines 72–73* and *137–138*). Each $G_t$ captures the pairwise interactions between objects at time step $t$, where each edge indicates whether an interaction exists between a pair of objects. Concretely, this can be represented as a binary adjacency matrix of size $N \times N$, where $N$ is the number of objects in the environment.
>
> To learn $G_t$, we introduce a **surrogate latent variable** $\mathbf{u}_t$ that parameterizes the distribution over interaction graphs. This modeling choice allows us to capture the fact that interactions and their underlying distributions may vary over time.
>
> Specifically, for each object pair $(i, j)$ at time $t$, we encode their latent states $s_t^i$ and $s_t^j$ using an RNN-based encoder to obtain a pairwise embedding:
> $$
> \mathbf{u}_t^{ij} = f\_{\text{enc}, \phi_u}(s_t^i, s_t^j)
> $$
> (as described in Equation (A2)).
>
> We then use these embeddings to infer the graph structure $G_t$ in one of two ways:
>
> 1. **Sampling edges** using a Gumbel-Softmax distribution over $\mathbf{u}_t$ (Equation A3), or
>
> 2. **Quantizing** $\mathbf{u}_t$ into a discrete codebook of interaction types (Equation A4), followed by decoding into a graph via:$G_t \sim g\_{\text{dec}}(e_z)$, where $e_z$ is the prototype vector.
>    (as shown in Equation (A5)).
>
> These mechanisms are designed to flexibly infer time-varying, discrete graph structures from continuous object states. More details can be found in Lines 146–151 and Appendix C.2. [**For clarity**] In the camera-ready version, we will move the above explanation into the main paper and expand it with additional details from Appendix C.2, as we have one extra page available.
>
> ---
>
> > Typos and color suggestion for Fig.6.
>
> **R2**:  [**For clarity**] We will correct all identified typos in the camera-ready version. For Fig. 6, we will use more distinguishable colors. For setrence at 112, we will make it as “We encode the observations using object-centric representation learning built on top of pre-trained models such as DINO-v2 [23] and R3M [24], which have been empirically shown to provide high-quality image understanding capabilities [26–28, 18, 29] and facilitate robotic manipulation tasks [19].”
>
> ---
>
> > How do you get the ground truth interaction structures G to get results shown Figure 6? Did you only do that for SpriteWorld, and that ground truth is not required unless you want to measure SHD? Also, how exactly is G represented?
>
> **R3**:  You are correct that ground-truth sequence of interaction graphs $G$ are only used in the SpriteWorld environment for evaluation purposes to compute the nSHD and we do *not* use it during training. During world model learning, we infer the estimated sequence of interaction graphs $\hat{G}$ in a self-supervised manner by maximizing the overall likelihood and optimizing the loss functions described in Section 3.1.
>
> As for obtaining the ground-truth $G$ in SpriteWorld, we derive it from visual inspection of the collected videos: if two objects are sufficiently close and exhibit abrupt changes in their trajectories, we label them as interacting. [**For clarity**] We will clarify this process and add the above explanation after Line 268 in the revised version.
>
> For representation, $G$ is a discrete binary adjacency matrix of size $N \times$, where $N$ is the number of objects. While the model may produce a continuous-valued matrix during optimization, we threshold it to obtain a binary matrix for evaluation. [**For clarity**] We will add this sentence after Line 73.
>
> ---
>
> > In the online stage, do you also update the world model with the new data you collect?
>
> **R4**:  For computational efficiency, and because our experiments show that the offline-trained world model is already robust given high-quality offline data, we do not update the world model during the online stage in our default setting. Please note that this is just an empirical choice, and it is feasible to incorporate online updates. To further investigate this, we conducted an ablation study comparing performance with and without online world model updates. The results are shown below:
>
> |Envs|FIOC (w/o Online Tuning)|FIOC (w/ Online Tuning)|
> |--|-|-|
> |Gym Fetch (Task 1)|0.95±0.03|0.96±0.02|
> |Gym Fetch (Task 2)|0.98±0.01|0.98±0.01|
> |Franka Kitchen (Task 1)|0.82±0.04|0.79±0.06|
> |Franka Kitchen (Task 2)|0.79±0.06|0.82±0.05|
> |i-Gibson (Task 1)|0.76±0.12|0.78±0.14|
> |i-Gibson (Task 2)|0.72±0.10|0.75±0.06|
> |Libero (Task 1)|0.81±0.11|0.83±0.08|
> |Libero (Task 2)| 0.74±0.09|0.71±0.06|
> |Push & Switch (Attri. Gen.)|0.91± 0.05|0.95±0.08|
> |i-Gibson (Attri. Gen.)|0.79±0.13|0.81±0.15|
> |Libero (Attri. Gen.)|0.76±0.14|0.68±0.09|
> |Push & Switch (Comp. Gen.)|0.86±0.10|0.82±0.12|
> |Libero (Comp. Gen.)|0.70±0.09|0.74±0.08|
> |Push & Switch (Skill Gen.)|0.81±0.06|0.82±0.08|
> |Franka Kitchen (Skill Gen.)|0.73±0.06|0.72±0.07|
>
> Results averaged over 5 seeds indicate that there is no significant difference, though we observe slight improvements in most environments for both single-task and generalization settings. [**For clarity**] We will include this comparison as an additional ablation study in the appendix, and clarify the experimental setting with the following description: “We also consider the case where online interaction data is used to further update the world model components during policy learning or planning. The results are given in Appendix D.” Additionally, we will emphasize in the main paper that “we fix the learned world model during policy learning and planning”, and will add this clarification after Line 150 to avoid confusion about whether the model is updated online by default.
>
> ---
>
> > Do you train the low-level policy fully inside the learned world model or using online observations?
>
> **R5**:  Thank you for the question. We train the low-level policy during the **offline training stage**, and then **fine-tune it during the online learning stage** using the newly collected interaction data. [**For clarity**] We will explicitly state at Line 167: “We learn the low-level policy during world model learning (Stage 1), and then fine-tune it with online data during Stage 2, where the policy is updated each time new interaction data becomes available.”
>
> We will provide the algorithm in the appendix (here is a sketch):
>
> **Offline World Model and Online Policy Learning Framework**
>
> ---
>
> **Stage 1: Offline Model Learning**
> For each $(o_t, a_t, r_t) \in \mathcal{D}$:
> - Encode observation: $\hat{o}_t \leftarrow p\_\text{pre-trained}(o_t)$
> - Infer latent state: $s_t^i \leftarrow q_\phi(s_t^i \mid \hat{o}_t)$
> - Factorize latent state: $s_t^i = (d_t^i, c_t^i)$
> - Learn reward model: $p_r(r_t \mid s_t, a_t)$
> - Learn transition model: $p_s(s_{t+1} \mid s_t, a_t, G_t)$
> - Infer interaction graphs: $p_u(G_t \mid s_t)$
> - Train interaction policy: $\pi^l(a_t \mid s_t, G_t^g)$
>
> ---
>
> **Stage 2: Online Policy Learning**
> For each environment rollout episode:
> - Receive current state $s_t$ from world model encoder with input environmental steps
> - Select goal graph: $G_t^g \sim \pi^h(G_t^g \mid s_t)$
> - Sample action from interaction policy: $a_t \sim \pi^l(a_t \mid s_t, G_t^g)$
> - Execute $a_t$ in environment, observe $r_t$, $o_{t+1}$
> - Update policies $\pi^h$, $\pi^l$ using collected data
>
> ---
>
>
> > In line 173, you say you only focus on subsets of objects with size of less than 2. Is that correct? Do you only focus on sets of single objects?
>
> **R6**:  Yes, at each step, we focus on a small subset of objects, typically one or two, to induce diverse interactions for them. This does not mean we only consider single-object interactions overall. Instead, we adopt an iterative strategy, where over multiple steps, we cover all objects by sequentially targeting different subsets. This is an empirical design choice that balances interaction diversity and sample efficiency.
>
> ---
>
> > FIOC considers two options for low-level policy: MPC and RL, and two options for graph inference: Variational and CIT. What version do you use in Table 2 results?
>
> **R7**:  We use **variational masks** for the results reported in Table 2. Specifically, for all methods (excluding those evaluated under nSHD), we adopt variational masks to infer the interaction structures. For downstream control, we apply **MPC** for the Gym-Fetch and Franka-Kitchen environments, and use **PPO** for LIBERO and iGibson tasks.
> We also conducted an ablation study to evaluate the impact of using MPC or PPO on all environments. The results of this ablation are shown below. The results indicate that using either PPO or MPC is feasible with similar final success rates. [**For clarity**] We will add this Table in Appendix D.
> |Environment|MPC|PPO|
> |-|-|-|
> |Fetch (Task 1)|0.95±0.03|0.96±0.03|
> |Fetch (Task 2)|0.98±0.01|0.92±0.05|
> |Kitchen (Task 1)|0.82±0.04|0.79±0.06|
> |Kitchen (Task 2)|0.79 ± 0.06|0.76±0.03|
> |i-Gibson (Task 1)|0.72±0.09|0.76±0.12|
> |i-Gibson (Task 2)|0.74±0.13|0.72±0.10|
> |Libero (Task 1)|0.73±0.05|0.81±0.11|
> |Libero (Task 2)|0.76±0.06|0.74±0.09|
>
> ---
> We hope these clarifications address your concerns. If so, we would be truly grateful if you would consider adjusting your rating accordingly. Thank you again for your time and efforts!

---

> > ### Comment · Reviewer_E9iB · 2025-08-03
> >
> > Thank you for answering my questions and for running additional experiments. I have a few more questions.
> >
> > 1. What is the dimension of $u$? Is it $u \in \mathbb{R}^{N \times N}$? Please add this to the paper if it's not there, that would help in understanding the method. And same for other variables you introduce. Having a clear understanding of variable dimensions and types would help in understanding what represents what.
> >
> > 2. > In line 173, you say you only focus on subsets of objects with size of less than 2. Is that correct? Do you only focus on sets of single objects?
> >
> > This line is still confusing to me, less than 2 means 1, so the phrase itself is confusing.
> > And I'm still confused as to what the high-level model is doing. My current understanding is that for a given state $S_t$ ,there's some interaction graph you can extract from it $G^t$. Then, the high-level policy $\pi_h$ should select the target interaction graph you want to reach next $G^g$. If we consider the Franka Kitchen environment, and let's say we're trying to move the kettle from the counter to the stove top, the sequence of intermediate subgoal graphs we'd consider is 1) graph where the arm and the kettle are interacting (arm is holding the kettle) and then 2) graph where kettle and the stovetop are interacting.
> > And by saying that you focus on a subset of objects with size of less than 2, you're saying that the delta between $G^t$ and $G^g$ should not be too big, meaning only one interaction between objects i, j, $G^g_{i, j} \neq G^t_{i,j}$, with the rest of the graph remaining unchanged.
> >
> > Is this understanding correct?
> > Also, is there any way to look into what the slots and interactions represent, or is it purely a black-box?
> >
> > 3. > FIOC considers two options for low-level policy: MPC and RL, and two options for graph inference: Variational and CIT. What version do you use in Table 2 results?
> >
> > Got it, please clearly state what version of your method you're using in each experiment in the paper.

---

> ### Author Response · Authors · 2025-08-04
> **Further Response (1/2)**
>
> Thank you very much for your feedback and follow-up questions. We address them point by point below.
>
> > What is the dimension of $u$? Is it $u \in \mathbb{R}^{N \times N}$? Please add this to the paper if it's not there, that would help in understanding the method. And same for other variables you introduce. Having a clear understanding of variable dimensions and types would help in understanding what represents what.
>
> For the two cases, we used different ways to sample $G_t$:
>
> 1. **Sampling edges** using a Gumbel-Softmax distribution over $\mathbf{u}_t \in \mathbb{R}^{N \times N}$ (Equation A3), which models the interaction logits between all object pairs. Here, the dimensionality of $\mathbf{u}_t$ is $N \times N$
>
> 2. **Quantizing** $\mathbf{u}\_t$ into a discrete codebook of interaction types (Equation A4), followed by decoding into a graph via:$
> G_t \sim g_{\text{dec}}(e_z),$
> where $e_z$ is the selected prototype vector. In our implementation, we set $\mathbf{u}_t$ equal to the quantized embedding. The 16-dimensional vector $e_z$ is then decoded into the parameters of Bernoulli distributions, from which we sample the final interaction graph $G_t$. Hence, here, the dimensionality of $\mathbf{u}_t$ is the same as $e_z$: 16.
>
> We will incorporate the dimensionality details, along with the description provided in our response to the first question in the rebuttal, into the main paper (after Line 151), while keeping the pointer to Appendix C.2 for further reference. To make it clear, we will also add a table for notations in Appendix C.
>
> > This line is still confusing to me, less than 2 means 1, so the phrase itself is confusing. And I'm still confused as to what the high-level model is doing. My current understanding is that for a given state $S_t$ ,there's some interaction graph you can extract from it $G^t$. Then, the high-level policy $\pi_h$ should select the target interaction graph you want to reach next $G^g$. If we consider the Franka Kitchen environment, and let's say we're trying to move the kettle from the counter to the stove top, the sequence of intermediate subgoal graphs we'd consider is 1) graph where the arm and the kettle are interacting (arm is holding the kettle) and then 2) graph where kettle and the stovetop are interacting. And by saying that you focus on a subset of objects with size of less than 2, you're saying that the delta between $G^t$ and $G^g$ should not be too big, meaning only one interaction between objects i, j, $G^g_{i, j} \neq G^t_{i,j}$, with the rest of the graph remaining unchanged.
>
> Yes, we should fix the main paper "less than 2" to "1 or 2". Thank for the pointer.
>
> For the rest of understanding, yes, you are correct. However, one clarification we would like to make is regarding the number of objects mentioned. For each subgoal interaction (i.e., the pair $(i, j)$), we first select one or two objects as candidates for $i$, and then sample $j$ conditioned on the chosen $i$. This hierarchical selection is designed for efficiency, as directly sampling from all $N \times N$ possible interactions would be computationally expensive.
>
>
> To clarify, we will add the following after Line 176 in camera-ready (we really like your example so we will also use it here, many thanks!!):
>
> ---
>
> To clarify the high-level planning process, we follow the perspective that each state $\mathbf{s}\_t$ is associated with an interaction graph $G^t$, and the final task corresponds to reaching a desired target graph $G^g$. $\pi_h$ selects a sequence of intermediate subgoal graphs that gradually transform $G^t$ into $G^g$ ($G^g_{i,j} \neq G^t_{i,j}$), where each subgoal graph differs from the previous one in only a single interaction. For example, in a task such as moving a kettle from the counter to the stove, the graph transitions might involve first enabling an interaction between the arm and the kettle, followed by an interaction between the kettle and the stovetop.
>
> To make the subgoal selection both tractable and structured, we do not sample directly from the full space of possible object interactions. Instead, at each decision point, we first identify a small subset of objects (typically 1–2) as primary candidates for initiating interaction changes. These candidates define the anchor object(s) $i$, and we then select a target object $j$ conditioned on $i$ to form the proposed subgoal interaction $(i, j)$. This scheme reduces the combinatorial action space and leads to more localized graph transitions. Note that the selected subset does not constrain the interaction to only occur between these objects; rather, it defines a focused region of the graph for subgoal exploration.
>
> ---

---

> ### Author Response · Authors · 2025-08-04
> **Further Response (2/2)**
>
> > Also, is there any way to look into what the slots and interactions represent, or is it purely a black-box?
>
> Yes, in the camera-ready version, we will include a figure visualizing the reconstructed slots. (Due to file upload constraints, we cannot include the image here, but it has already been prepared and will be added.)  For interactions, we can find the list of object pairs $(i, j)$ that are activated at each high-level step. This will clearly indicate which objects are interacting or being focused on during each subgoal transition.
>
> > Got it, please clearly state what version of your method you're using in each experiment in the paper.
>
> We will insert the following clarification after Line 233 in the camera-ready version: "We use *variational masks* for the results in Table 2. Specifically, for all methods (excluding those evaluated under nSHD), we adopt variational masks to infer the interaction structures. For downstream control, we apply *MPC* for the Gym-Fetch and Franka-Kitchen environments, and use *PPO* for LIBERO and iGibson tasks." We will put further ablations (given in the rebuttal) in Appendix D.
>
> ---
>
> We would like to once again express our sincere gratitude to the reviewer for the thoughtful review. Your suggestions have helped strengthen the paper. We hope the above responses have addressed your further questions, and please do not hesitate to reach out if there are any remaining concerns.

---

> > ### Author Response · Authors · 2025-08-06
> > **Thank you for your additional feedback – Follow-up on our responses**
> >
> > Dear Reviewer E9iB,
> >
> > Thank you for your insightful comments and additional feedback. We have provided further responses above to address the new questions you raised. Please feel free to let us know if any concerns remain or if further clarification is needed.
> >
> > Thank you again for your valuable comments and for your time and effort in reviewing our work. We truly appreciate your insights.
> >
> > Best,
> >
> > Authors of Submission7735

---

> > > ### Comment · Reviewer_E9iB · 2025-08-06
> > >
> > > Thank you for answering my questions and helping me understand the method, most of my concerns are addressed.
> > > Conditional on you implementing the changes we discussed to improve the presentation clarity, I raise my score to 4.

---

> > > > ### Author Response · Authors · 2025-08-06
> > > > **Thank you!**
> > > >
> > > > Thank you so much for the positive feedback and for adjusting your consideration. We are glad we could address your concerns. The comments and questions were all insightful for improving the work, and we truly appreciate the time and effort you've dedicated! We will make sure to implement the content changes outlined in the rebuttal in the camera-ready.

---

### Note · Authors · 2025-08-12

We thank all reviewers for their time and effort in the review and discussion phases, and the AC for maintaining an insightful and rigorous process. Here, we briefly summarize the rebuttal and discussion, and outline the points to be included in the camera-ready version.

First, we appreciate all reviewers’ recognition of our motivation, model design, and evaluation, and are pleased that our rebuttal addressed your concerns, as also reflected in the adjusted ratings. We provided clarifications on technical details and additional evaluations complementing the main results, with *exact sentences and placement* specified in the rebuttal. Below, we summarize key experiments (we've done in the rebuttal) and planned revisions to be reflected in the camera-ready version.

**New ablations** we've done in the rebuttal along with their placement in the camera-ready version:

- Two-stage learning: Updating the world model during policy learning (`R4` to E9iB, Appendix D.2);
- World model: More structured learning methods (`R4`, to VDhr, Appendix D.1); Number of slots in SA (`R5` to gBUX, Appendix D.1);
- Policy learning: MPC vs PPO across tasks (`R7` to E9iB, `R2` to Kuio, Appendix D.2); Pre-trained models for Dreamer & TD-MPC (`R5` to VDhr, Appendix D.2); Mixing ratio of model rollouts vs replay buffer (further discussion w/ VDhr, Appendix D.2); Longer horizons on Kitchen (`R7` to Kuio, Table 2).

**Other revisions**: Elaborating on graph learning (specified in `R1` to E9iB, `R2–R3` to VDhr, ~4–5 lines); clarifying PPO/MPC usage (`R7` to E9iB, `R2` to Kuio, ~1–2 lines); expanding on number of objects in high-level policy (specified in  Further Response to E9iB, ~3-4 lines); adding the two-stage learning algorithm (to be put in Appendix C, algorithm given in `R5` to E9iB, `R3` to gBUX); and adding visualizations of reconstructed slots and disentanglement to complement LPIPS (Table 1) and probing (Fig. 5).  Together with the other detailed clarifications specified in the rebuttal (*with exact sentences and their placement in the rebuttal*), all can be included **within the extra page allowed**.

We believe that all the new evaluations and changes to text will further improve clarity and provide stronger support for our main findings. These revisions are fully executable and we will make sure these to be incorporated into the camera-ready version. Thank you again for your time and constructive feedback, and we hope these updates address all remaining concerns.

---

### Decision · Program_Chairs · 2025-09-17

**Decision:**

Accept (poster)

**Comment:**

This paper proposes FIOC-WM, a two-stage framework for object-centric model-based reinforcement learning. It first learns a structured world model by factorizing observations into object-centric representations with static and dynamic attributes and modeling inter-object interactions as sparse, time-varying graphs. A hierarchical policy then plans over these interactions: a high-level policy sets subgoals, while a low-level policy executes interactions to optimize task rewards. Experiments on robotic manipulation and simulated environments show improved policy performance, better generalization to unseen object combinations, and enhanced predictive fidelity compared to strong baselines including DreamerV3, TD-MPC2, DINO-WM, and prior object-centric RL methods.

Reviewers praised the principled object-centric factorization, hierarchical policy design, and strong empirical results, highlighting the method’s compositionality and generalization advantages. At the same time, concerns were raised about presentation clarity, missing qualitative visualizations, experimental details, and motivation for certain design choices. The rebuttal addressed some of these concerns, providing clarifications and committing to substantial revisions in the final version.

After evaluating the reviews and rebuttal, I recommend ACCEPT (Consensus Reached among Reviewers). The authors are expected to implement the promised improvements, including enhanced clarity and complete experimental details, in the final version to fully address reviewer concerns.